# Rapid discovery of high-affinity antibodies via massively parallel sequencing, ribosome display and affinity screening

Benjamin T. Porebski [1], Matthew Balmforth [1,4], Gareth Browne[2], Aidan Riley[2], Kiarash Jamali [1], Maximillian J. L. J. Fürst [1,5], Mirko Velic[3], Andrew Buchanan [2], Ralph Minter [2,6], Tristan Vaughan[2] & Philipp Holliger [1]✉

Developing therapeutic antibodies is laborious and costly. Here we report a method for antibody discovery that leverages the Illumina HiSeq platform to, within 3 days, screen in the order of $10^8$ antibody–antigen interactions. The method, which we named 'deep screening', involves the clustering and sequencing of antibody libraries, the conversion of the DNA clusters into complementary RNA clusters covalently linked to the instrument's flow-cell surface on the same location, the in situ translation of the clusters into antibodies tethered via ribosome display, and their screening via fluorescently labelled antigens. By using deep screening, we discovered low-nanomolar nanobodies to a model antigen using $4 \times 10^6$ unique variants from yeast-display-enriched libraries, and high-picomolar single-chain antibody fragment leads for human interleukin-7 directly from unselected synthetic repertoires. We also leveraged deep screening of a library of $2.4 \times 10^5$ sequences of the third complementarity-determining region of the heavy chain of an anti-human epidermal growth factor receptor 2 (HER2) antibody as input for a large language model that generated new single-chain antibody fragment sequences with higher affinity for HER2 than those in the original library.

Massively parallel assays provide the ability to enormously increase both the throughput and speed of data generation in the biomedical sciences, and have proven key to the discovery of antibody, peptide and aptamer leads and enzymatic catalysts[1–3]. Although methods of diversification at the level of high-throughput DNA oligonucleotide synthesis are highly developed[4], and various selection strategies such as phage, yeast and ribosome display[5] are able to process large combinatorial (poly)peptide repertoires, these still sample only a fraction of the possible sequence space. Furthermore, all selection methods (to different degrees) suffer from inherent and inescapable additive

biases that hinder discovery. Also, such selections are generally conducted 'in the blind', with little or no overall a priori information on the likelihood of successful outcomes.

Next-generation sequencing (NGS) can provide information on the distribution and enrichment of genotypes during selection experiments, but multiple studies suggest that repertoire-selection experiments, such as phage display, are prone to biases and to inefficient enrichment[5,6] owing to varying levels of efficiency of protein expression, display and folding, and to fitness effects on the host organism. Therefore, the genotype distribution, abundance and

[1]MRC Laboratory of Molecular Biology, Cambridge, UK. [2]Biologics Engineering, AstraZeneca, Cambridge, UK. [3]Docklands, Victoria, Australia. [4]Present address: UCB, Slough, UK. [5]Present address: Groningen Biomolecular Sciences and Biotechnology Institute, University of Groningen, Groningen, the Netherlands. [6]Present address: Alchemab Therapeutics, London, UK. ✉e-mail: ph1@mrc-lmb.cam.ac.uk

enrichment obtained from sequencing data only provides an imperfect proxy for function and for the global phenotype distribution of a biomolecular repertoire.

Owing to these limitations, and the desire to obtain a more reliable global picture of genotype-to-phenotype correlations, numerous high-throughput screening methods have been developed; however, the majority of screening approaches are limited in scope, scale and information output. Isolated screening (one clone per compartment) does not easily scale, even with robotics or microfluidics, and as a result it is expensive to determine the sequence composition of each clone, and is often only done for the identified hits[7]. Array-based assays, where a known sequence is printed, synthesized or captured in a defined position, allow for the coupled measurement of sequence and function and are powerful, but remain limited in scale[7–11].

A potentially transformative approach seeks to merge NGS directly with functional screening. NGS technologies on the Polony[12] and Illumina[13] platforms rely on extreme parallelization by sequencing clonal DNA from randomly arrayed DNA clusters. Both platforms have been leveraged either directly or through barcoding for the parallel interrogation of hundreds of thousands of DNA–protein, RNA–protein and protein–protein interactions[14–20].

Here we present 'deep screening', a method that leverages the Illumina HiSeq platform to array, sequence and screen antibody libraries. Deep screening involves the clustering and sequencing of antibody libraries at the DNA level, followed by the conversion of Illumina flow-cell DNA clusters into complementary RNA clusters that are covalently linked to the flow-cell surface in the same location. RNA clusters can either be interrogated directly or preferentially translated into proteins and tethered via ribosome display. The apparent equilibrium-binding affinities and dissociation kinetics of the displayed proteins to a fluorescently labelled target ligand can then be determined at scale, with the entire process being performed on the HiSeq platform. Focussing here on antibody discovery, we show the deep screening of yeast display pre-selected libraries of synthetic camelid single-domain antibody fragments (VHH nanobodies) and of unselected synthetic human single-chain antibody fragment (scFv) libraries, with the discovery of high-affinity (low nanomolar to mid picomolar) binders directly from global antigen-binding data, accelerating high-affinity antibody-lead discovery from months to 2–3 days. We also show the utility of deep-screening datasets as input for a machine learning (ML) model trained on antibody–antigen interactions for the rapid generation of new high-affinity antibody-lead sequences that exceed the performance of those present in the original library.

## Results

### Implementation of deep screening

The Illumina HiSeq 2500 is an NGS platform that operates on a highly integrated instrument with a flow cell comprising up to $2 \times 10^9$ clonal DNA clusters. These clusters are generated in situ from individual, single-stranded DNA template molecules by a process called bridge amplification. Individual clusters typically comprise an array of approximately 1,000 DNA molecules in a roughly 1-µm-diameter spot[13]. Once arrayed, clusters are sequenced in parallel using Illumina's sequencing-by-synthesis technology, yielding sequences and their physical x–y coordinates as an output.

Development of ultra-high-throughput antibody screening on this platform faced multiple technical challenges as described below. To implement screening of protein interactions at the localization of the sequenced clusters required the development of new methodologies to convert DNA clusters into RNA and then protein clusters. To this end, we leveraged the efficient primer-dependent RNA polymerase activity of the engineered *Thermococcus gorgonarius* DNA polymerase TGK[21] to convert post-sequencing DNA clusters into RNA clusters. Specifically, we exploit the paired-end turnaround process (a standard process on the Illumina platform to regenerate the anti-sense strand after sequencing and convert it to the sense strand) to perform DNA-templated RNA synthesis, whereby the surface-linked Illumina primer (P5) is repeatedly extended by TGK-mediated RNA synthesis. Once RNA synthesis is complete, the DNA template is removed, creating single-stranded RNA clusters that are covalently linked to the flow-cell surface via the P5 primer (Fig. 1). Our approach differs from similar strategies implemented on the Illumina GenomeAnalyzer[17] in that flow-cell-bound primers do not need to be modified to enable RNA synthesis, and removal of DNA templates does not require DNase I treatment. Deep screening also differs from an approach implemented on the Illumina MiSeq platform[18], which uses a stalled *Escherichia coli* RNA polymerase to non-covalently link the transcribed mRNA to double-stranded DNA clusters, in that deep-screening-displayed mRNAs are covalently linked to the flow-cell surface, enabling enhanced display stability and flexibility in assay reagents and temperatures.

Next, we developed a workflow to translate RNA clusters and stably display the resulting polypeptides on the flow-cell surface. As 5'-tethered RNA clusters are vulnerable to nuclease degradation, we used the reconstituted PURExpress in vitro translation (IVT) system[22]. We specifically used PURExpress ΔRF123, −T7 RNAP—which lacks all release factors (RF-1, -2 and -3) and T7 RNA polymerase—in conjunction with a flow-cell-tethered RNA construct that comprises the desired open reading frame (ORF) preceded by a 5' untranslated region composed of an $N_{28}$ random DNA sequence segment serving as a unique molecular identifier (UMI or barcode) and a translation initiation signal (Shine−Dalgarno (SD)). After the ORF, we introduced a 3' extension sequence encoding a 40 amino acid (AA) polypeptide sequence (to prevent partial retainment of the displayed protein within the ribosomal exit tunnel) and two stop codons for ribosome stalling (Fig. 1). Stalled mRNA−ribosome−nascent polypeptide complexes are stable for several days at ambient temperature in a high magnesium buffer, during which the flow-cell array of protein clusters with known sequences or UMIs and coordinates can be interrogated in a variety of functional assays, such as antigen binding.

In summary, the deep-screening workflow (Fig. 1) involves (1) sequencing of $N_{28}$ UMI barcodes, (2) conversion of DNA into RNA clusters, (3) IVT of the RNA clusters into protein clusters, and (4) interrogation of protein clusters for equilibrium binding and dissociation for a target ligand. Clone ranking and in situ affinity and kinetic data are calculated from raw flow-cell images by associating UMIs with fluorescence intensities (FIs) at different equilibrium ligand concentrations and during the washing steps, yielding apparent dissociation constants ($K_D^{app}$) and kinetic off rates ($k_{off}^{app}$) for each UMI. Next, standard sequencing of the library is performed on the HiSeq 2500 (or any other sequencing instrument) using a new flow cell to link ORF sequences with UMI barcodes, and therefore binding data (Fig. 1c). Depending on the number and length of the sequence regions to be sequenced, a complete deep-screening experiment (including data processing and hit selection) can be completed in ≤3 days (Fig. 1d).

### Nanobody discovery with pre-selection

For proof of concept, we first explored deep screening of a commercially available yeast surface display nanobody (VHH) library[23]. We performed two rounds of positive and negative selection by magnetic-activated cell sorting (MACS), followed by either an additional round of MACS or by fluorescence-activated cell sorting (FACS) for binding to a model antigen (hen egg lysozyme (HEL)) before deep screening the selection outputs on a flow cell (Fig. 2a and Supplementary Fig. 1) using 12-fold redundancy of any given UMI. Redundancy reduces noise in the binding datasets, improves unambiguous hit identification and minimizes false positives, although at the expense of reducing screening depth. Together with the diversity losses incurred by bottlenecking and clustering of the same library on two separate flow cells (for VHH sequencing), this yields a theoretical maximum of $2.5 \times 10^7$ UMIs. We did not reach this but recorded $3 \times 10^6$ $UMI_{FACS}$ and

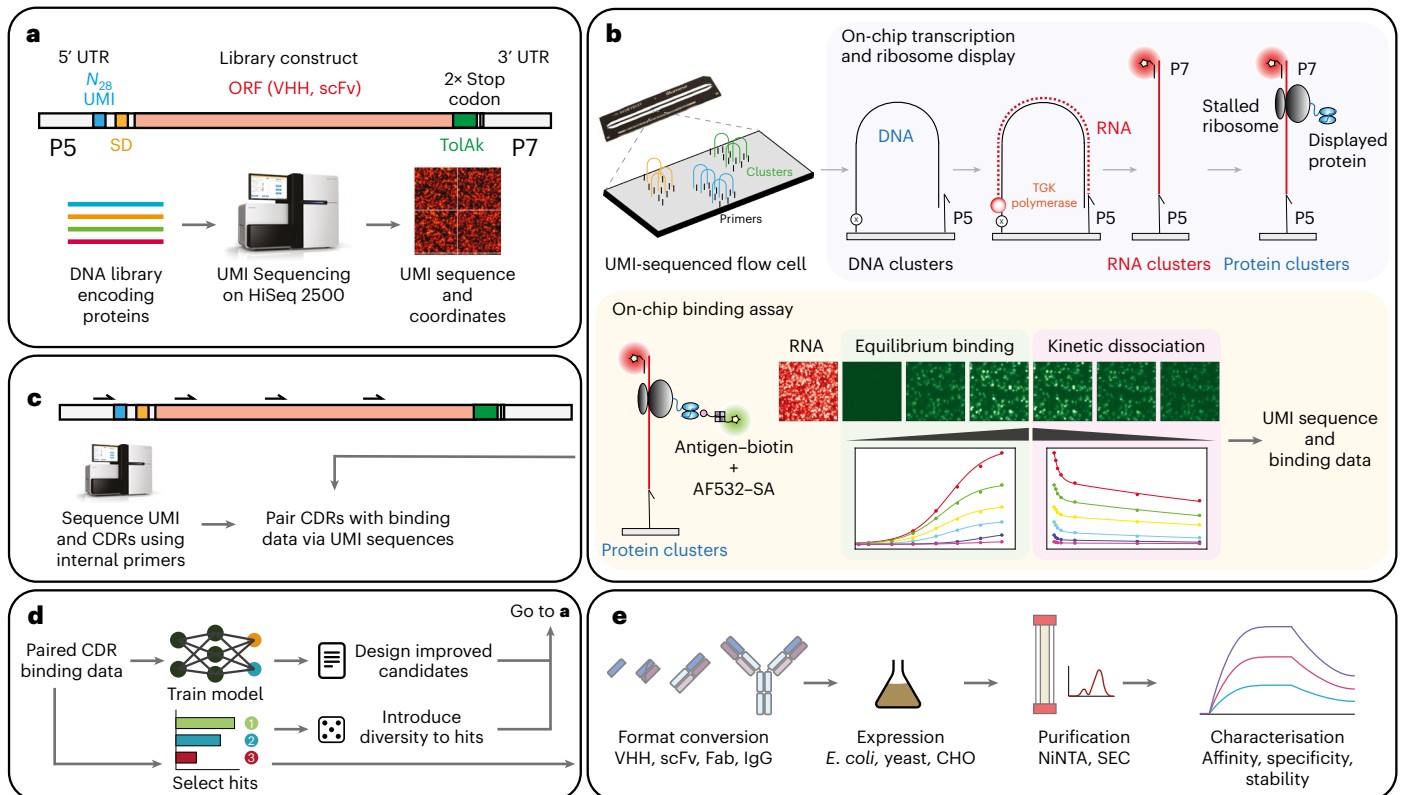

**Fig. 1 | Deep-screening workflow. a**, Antibody library preparation involves the addition of 5′ and 3′ untranslated regions (UTRs) that flank the library protein-coding region. The assembled library is then clustered and the $N_{28}$ UMI sequenced on a HiSeq 2500, which reports the UMI sequence and its physical $x$–$y$ coordinates on the flow cell. **b**, In deep screening, DNA clusters are converted into RNA clusters using engineered polymerase TGK[21] and the DNA template is removed. The RNA clusters are labelled with a complementary Atto 647N-labelled oligonucleotide before IVT into protein (antibody) clusters. Cluster binding is determined by equilibrium binding of an increasing concentration of biotinylated antigen and AF532-labelled Streptavidin (SA), followed by kinetic dissociation from the highest antigen concentration. **c**, If the binding assay reports hits within the library, a second sequencing experiment is performed to determine the UMI and CDRs with internal sequencing primers. CDRs are then paired with binding data using the common UMIs between the two experiments. **d,e**, Paired CDR–binding data is analysed for hits and/or a ML model is trained to predict hits, which can be used to generate libraries for subsequent rounds of deep screening (**d**) or short-list hit candidates for characterization via conversion into an appropriate antibody format, expression, purification (using methods like nickel-nitrilotriacetic acid (NiNTA) resin and size exclusion chromatography (SEC)) (**e**).

$4.4 \times 10^5$ UMI$_{MACS}$. To rank hits by apparent HEL binding affinities ($K_D^{app}$), we performed an in situ equilibrium-binding affinity titration comprising escalating concentrations of HEL (up to 300 nM), followed by measurement of dissociation rates whereby the rate of FI decay during washing was used to calculate an apparent dissociation constant ($k_{off}^{app}$). Data analysis identified 3,687 (FACS) and 1,479 (MACS) unique, putative hits with a mean integrated-cluster FI signal exceeding the library mean FI (FI$_{mean}$) by at least a factor of 2 (at the highest concentration of antigen tested (300 nM HEL, FI$_{mean}$ = 173.79, selected hit threshold of 347.58 FI units). As we empirically discovered in later screening experiments, a less-stringent hit FI threshold of 1.5-fold over the library mean is sufficient to avoid false positives. We next performed library sequencing to link VHH complementarity-determining region (CDR) 1–3 sequences (VHH genotypes) to their equilibrium-binding signals and dissociation rates ($K_D^{app}$ and $k_{off}^{app}$; VHH phenotypes), obtaining 379,300 (MACS) and 39,900 (FACS) unique CDR combinations (Fig. 2b), from which we identified 47 (MACS) and 53 (FACS) unique putative VHH hits.

However, reliable hit identification rests on the conjecture that a high peak FI and/or equilibrium-binding signal ($K_D^{app}$) correlates with 'true' high-affinity binding ($K_D$). To test this hypothesis, we characterized 20 clones (M1–M19 and M23) from the round 3 (R3) MACS and 10 clones (F1–F10) from the R3 FACS screens, spanning a wide range of observed FI signals, $K_D^{app}$s and abundances for characterization (Extended Data Fig. 1a). At the same time, we picked 96 random colonies

from the R3 MACS selection for colony PCR and Sanger sequencing, yielding 25 unique CDR sequences (see Source data for Extended Data Fig. 1), which included 4 clones already selected from the MACS or FACS libraries (F1, M18, F6 and F3) as these were highly abundant or enriched clones. From the remaining 21 sequences, we selected 8 clones (C1–C8) for a total of 38 variants for measurement of binding kinetics by bio-layer interferometry (BLI; Fig. 2d,e, Extended Data Fig. 1 and Supplementary Figs. 3–5). We identified three VHH hits with low-nanomolar $K_D$s (M5, $1.9 \times 10^{-8}$ M; M6, $1.42 \times 10^{-8}$ M; M15, $9.81 \times 10^{-9}$ M), and nine clones with lower affinity $K_D$s, ranging from 20 nM to 100 nM, including two from the randomly picked colonies (C1 and C2) (see Source data for Fig. 2d,e, Extended Data Fig. 1 and Supplementary Figs. 2–5). Plotting $K_D$ values derived from BLI measurements against FI$_{mean}$ values determined in deep screening resulted in a Spearman's rank correlation coefficient ($r_s$) of −0.697, $P < 0.001$ at 300 nM HEL (the highest concentration tested in the binding assay and the condition that shows the greatest separation of FIs; Extended Data Fig. 1b). We also plotted BLI $K_D$ values against the deep-screening-derived $K_D^{app}$ values, which revealed $r_s = 0.574$, $P = 0.0014$ (Extended Data Fig. 1c). In both cases, we set clones with no determined binding affinity as $>1 \times 10^{-5}$ M as we could not meaningfully detect any binding beyond this point. Reformatted as a binary classification of hit versus non-hit, this yielded a weighted F1 score of 0.79 and receiver operating characteristic area under the curve of 0.76, which is shown in Extended Data Fig. 1d. As our

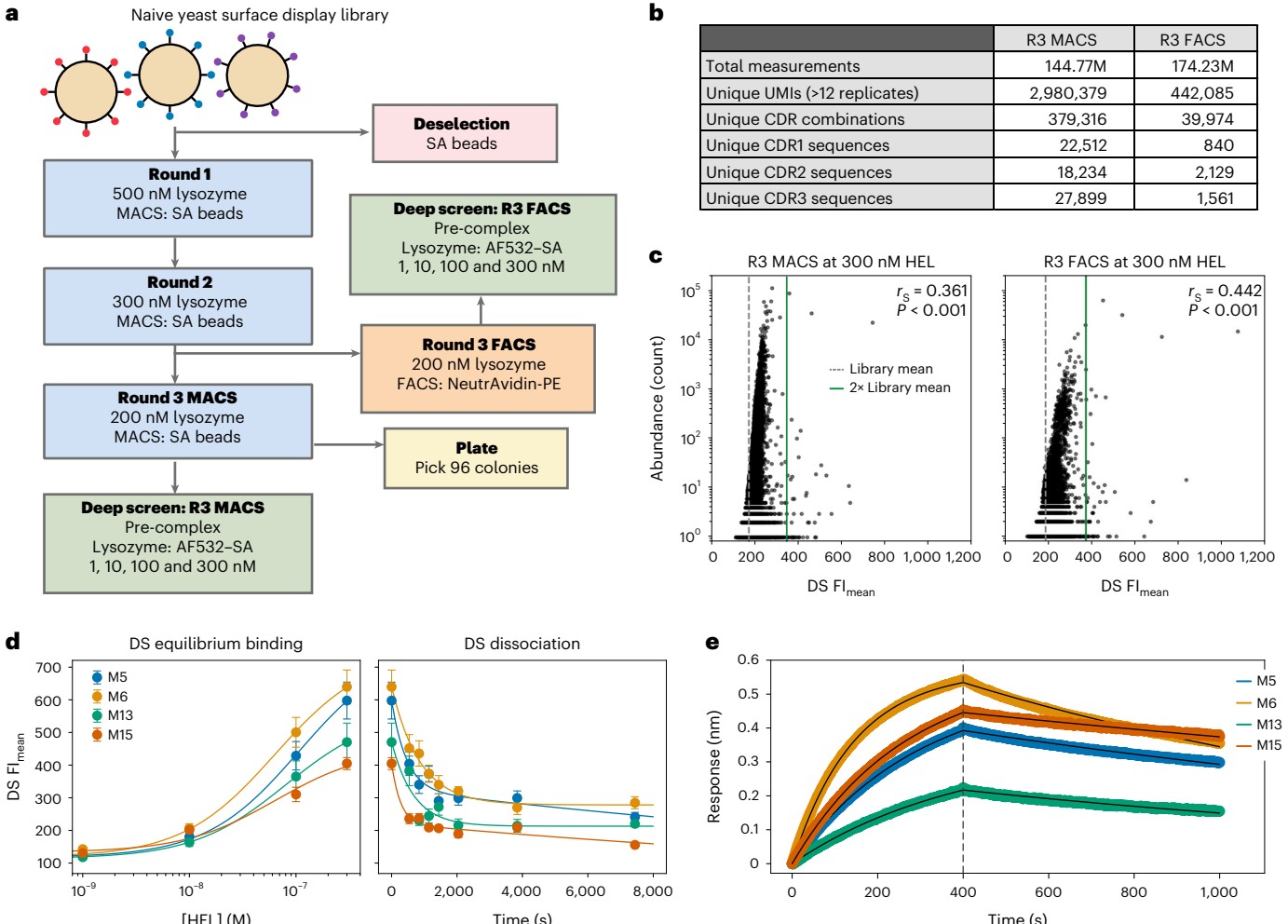

**Fig. 2 | Deep screening of a yeast-display-pre-selected VHH library.**
**a**, Workflow of VHH yeast display selections. **b**, Library statistics, showing the total number of clusters or reads, the number of barcodes or UMIs with 12 replicates and number of unique CDR combinations in the protein space. **c**, Abundance versus deep-screening (DS) $FI_{mean}$ of unique CDRs at 300 nM HEL from the R3 MACS and R3 FACS libraries. The library mean intensity is shown as a grey dashed line, and a solid green line shows the hit threshold of 2× the library

background. $r_S$ values of 0.361 and 0.442, respectively, show a poor correlation between abundance and deep-screening binding intensities. **d**, Deep-screening equilibrium-binding and kinetic dissociation curves for clones M5, M6, M14 and M15. Error bars are s.e.m. and $n \geq 12$ technical replicates of a given UMI. **e**, BLI kinetics at 50 nM of the same 4 clones against a HEL–biotin-loaded SA tip. The grey dashed line is denoting the separation of the association (left) and dissociation (right) phases collected during BLI kinetics measurements.

anti-HEL deep-screening experiments did not include a streptavidin (SA) screen, SA-binding VHHs would probably be mis-identified as binders in deep screening, but fail to generate detectable HEL binding by BLI, which used an SA-only tip for referencing. This together with potential failure of bacterial VHH expression are the likely reasons for the occurrence of false positives in this experiment. Nevertheless, this experiment provided proof-of-principle for deep-screening as an approach for the rapid identification of low-nanomolar VHH binders from pre-selected yeast display repertoires and indicated that both FI and $K_D^{app}$ values serve as adequate correlates for ranking VHH clones.

This experiment also showed a key advantage of deep screening over alternative strategies, in that deep-screening datasets provide a global and granular overview of library performance and enable a detailed analysis of the antibody discovery process. For both MACS and FACS selection, we observed a poor correlation ($r_S = 0.361$ for MACS; $r_S = 0.442$ for FACS) between CDR abundance and peak FI (as a correlate of affinity; Extended Data Fig. 1b) (Fig. 2c). This result suggests that both the MACS and (to a lesser extent) FACS selections enriched high-affinity clones inefficiently, presumably due to well-known inefficiencies of yeast display, including biases such as non-specific binding

and clonal variances in growth (that is, host toxicity), expression, folding or display. Deep screening can bypass some of these inefficiencies, as shown by the isolation of rare high-affinity binders from both selections (for example, poorly enriched high-affinity VHH-M5, -M6 and -M15 clones, with just 3, 11 and 145 UMIs in $2.9 \times 10^6$ screened), that would have been challenging to discover in the absence of further rounds of selection or laborious microplate screening of thousands of colonies.

## scFv antibody discovery without pre-selection

Having validated our approach using VHH nanobody discovery, we sought to explore whether deep screening could enable antibody discovery directly from an unselected library to avoid the enrichment biases observed during bulk selections. Specifically, we sought to explore direct discovery of high-affinity scFvs against a clinically relevant target. As our starting point, we chose IL70001, a human scFv antibody lead candidate. IL70001 had been previously isolated by phage display as a lead with micromolar affinity and a half-maximal inhibitory concentration ($IC_{50}$) of 7.3 μM against human interleukin-7 (huIL-7), a potential drug target implicated in autoimmune and allergic inflammatory disease[24-26] (Supplementary Figs. 7 and 8). From IL70001,

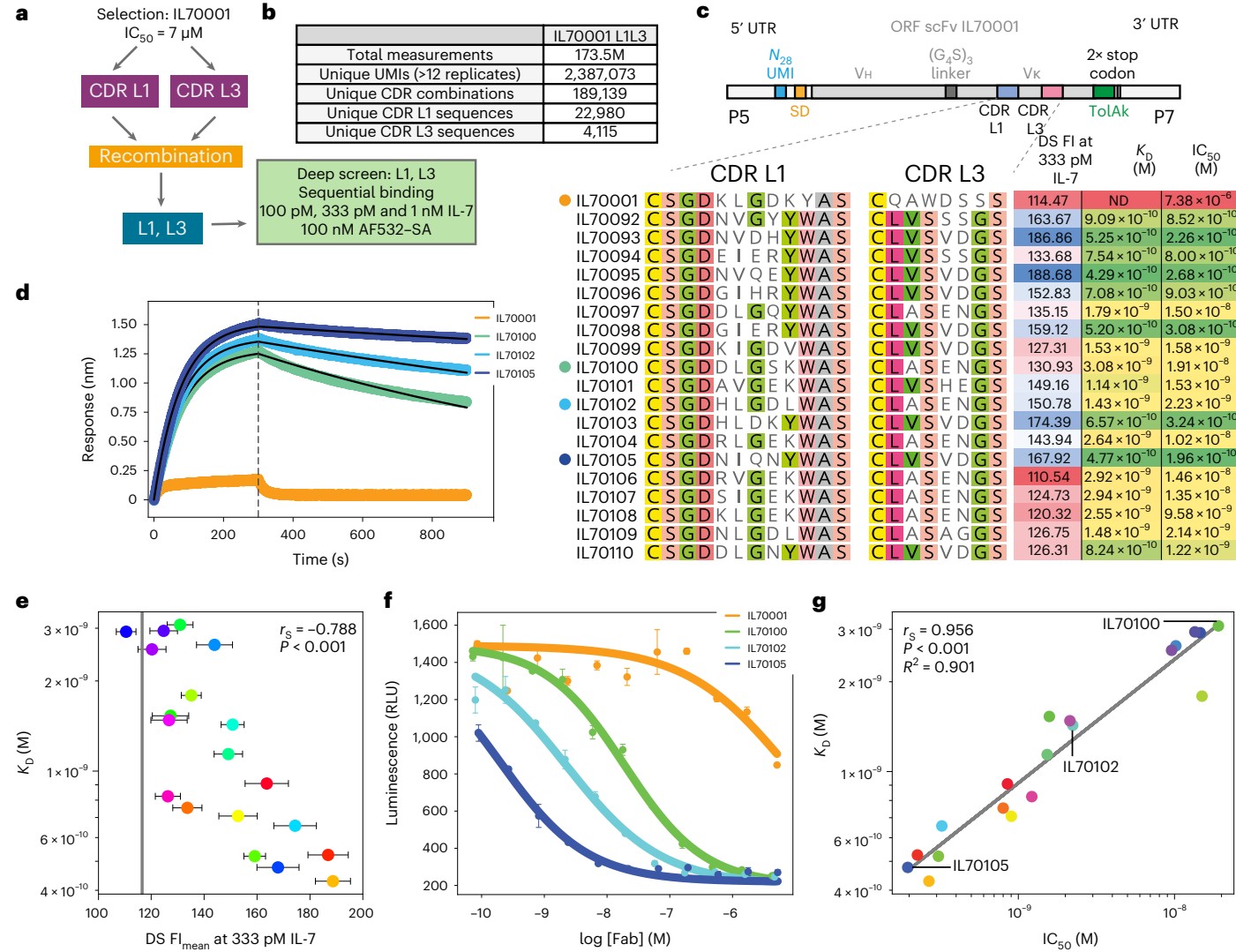

**Fig. 3 | Deep screening of an unselected scFv library. a**, Overview of the direct affinity maturation experiment from an unselected CDR L1, L3 affinity maturation library. **b**, Library statistics of the unselected L1L3 library from deep screening. **c**, IL70001 and the top-19 clones, showing CDR L1 and CDR L3 sequences, raw deep-screening intensities at 333 pM huIL-7, BLI-fitted $K_D$s and IL-7R IC50s. **d**, BLI kinetics at 50 nM of IL70001, IL70100, IL70102 and IL70105 Fabs against a huIL-7-loaded SA tip. The grey dashed line is denoting the separation of the association (left) and dissociation (right) phases collected during BLI kinetics measurements. **e**, Deep-screening FImeans of the top-19 clones at 333 pM huIL-7 plotted against fitted BLI $K_D$s. Error bars are s.e.m. and $n \geq 12$ technical replicates

of a given UMI. The grey vertical line shows the mean library intensity at 333 pM huIL-7. **f**, TF-1 STAT5 IL-7Rα and IL-7Rγ luciferase inhibition assay, showing mean signal from IL70001, IL70100, IL70102 and IL70105 as a representative range of the assay. All inhibition assay curves are shown in Supplementary Fig. 7. Error bars are the minimum and maximum observations, $n = 2$ technical replicates. **g**, Plotting BLI-fitted $K_D$s against IC50 reveals a strong, linear correlation between affinity and inhibition ($r_S = 0.956$, $R^2 = 0.901$). As IL70001's $K_D$ is probably considerably larger than 50 nM, the maximum response measured and speed of the on and off rates was insufficient for an accurate fit of the $K_D$. ND, not determined.

an affinity maturation library had been constructed by diversification of both Vκ light-chain CDRs L1 and L3, and we subjected this unselected library directly to deep screening (Fig. 3a).

Deep screening and CDR L1 and L3 sequencing yielded $1.7 \times 10^8$ measurements comprising $2.4 \times 10^6$ unique UMIs (with ≥12 replicates), and $1.9 \times 10^5$ unique CDR combinations in protein space (Fig. 3b). Due to huIL-7's tendency to aggregate at higher concentrations, we only collected equilibrium-binding data up to 1 nM of huIL-7, which resulted in 173 unique, potential hit UMIs. Sequencing of putative hits showed a general convergence of CDR L3 loop sequences (despite the large diversity of the input library), while retaining considerable diversity in the central region of CDR L1, presumably reflecting the larger contribution of CDR L3 to the huIL-7 paratope. Therefore, we selected a subset (top-19 clones as

judged by equilibrium-binding signal at 1 nM huIL-7; Supplementary Fig. 6a,b) plus the parental clone, IL70001, for characterization in greater depth (see Source data for Fig. 3c). These were re-cloned as antibody antigen-binding fragments (Fabs) (to avoid potential pitfalls in affinity measurements due to scFv multimerization), expressed and purified from Chinese hamster ovary (CHO) cells, and binding kinetics were measured by BLI at 50 nM of each Fab. This showed that all 19 anti-huIL-7 Fabs have $K_D$ values ranging from 3 nM up to 429 pM, representing an up to 2,300-fold improvement over the parent clone (see Source data for Fig. 3c,d and Supplementary Fig. 7), assuming a 1 μM $K_D$ for the parent. Again, we observed a strong correlation between equilibrium-binding signal and BLI measured affinities ($r_S = -0.788$; Fig. 3c,e and Supplementary Fig. 6b), even when switching antibody formats from scFv to Fab.

IL-7's role in autoimmune and allergic inflammatory diseases is mediated by the IL-7 receptor (IL-7R) (refs. 24–26). Therefore, we sought to assess whether our high-affinity Fab hits could inhibit huIL-7R signalling through huIL-7 sequestration using a TF-1 STAT5 IL-7Rα and IL-7Rγ luciferase cell-based reporter assay. Indeed, we observed an average 10,000-fold increase in inhibition potency ($IC_{50}$) over IL70001, with clone IL70105 yielding a 37,000-fold improvement (see Source data for Fig. 3f and Supplementary Figs. 8 and 9) with an excellent correlation between affinity and inhibition potency ($r_s = 0.956$, $R^2 = 0.901$; Fig. 3g).

This shows that deep screening can rapidly identify multiple high-picomolar-affinity antibodies against a therapeutically relevant drug target directly from an unselected library. Bypassing selection, deep screening delivers a major increase in discovery speed and provides a direct route to high-affinity (picomolar) antibodies without the need for pre-selection or pre-enrichment steps and their associated biases and inefficiencies. Furthermore, isolated Fab clones showed universally favourable general properties and developability indicators, such as good expression yields (0.25–0.6 mg ml$^{-1}$ of culture) and excellent monomericity (12 of 19 clones showed ≥98% monomeric fraction) as per high-performance size exclusion chromatography (see Source data and Supplementary Fig. 10).

## scFv antibody discovery augmented by ML

Deep screening produces large, internally consistent datasets linking antibody sequence to function. Next, we sought to explore whether such datasets could be leveraged for supervised machine learning (ML) to enable an even wider exploration of CDR sequence space and further accelerate high-affinity antibody discovery. As a target, we chose human epidermal growth factor receptor 2 (HER2; ERBB2), a cell-surface protein tyrosine kinase that is overexpressed in 30% of breast[27,28], ovarian[29,30] and lung cancers[31]. HER2 is also the target of the highly effective therapeutic antibody trastuzumab, which has a reported binding affinity ($K_D$) between 0.1 nM and 0.5 nM (refs. 32–34). To benchmark scFv display and HER2 binding on the flow cell, we selected a well-characterized panel of five anti-HER2 scFvs (G98A, C6.5, ML3-9, H3B1 and B1D2+A1) with reported binding affinities ($K_D$) between 320 nM and 15 pM (ref. 35) (Extended Data Fig. 2a,b). We observed a generally correct ranking (Extended Data Fig. 2a,c), with the caveat that high-affinity clones ML3-9, H3B1 and B1D2+A1 and similarly low-affinity clones G98A and C6.5 were only weakly separated by $K_D^{app}$ (Extended Data Fig. 2a). Directly comparing peak FI values at 100 nM HER2, the clone separation and ranking was more in line with reported $K_D$ values, but vulnerable to differences in expression and folding levels, as illustrated by trastuzumab showing a substantially higher peak FI relative to the affinity panel clones (Extended Data Fig. 2c). Thus, although both $K_D^{app}$ and peak FI provide principally correct affinity rankings, the caveat is that deep-screening data are a complex mixture of flow-cell display efficiency and binding that a simple hill equation fit cannot fully represent. At the same time, peak FI clearly captures desirable features beyond antigen affinity, such as relative efficiencies of functional antibody expression and folding.

We chose the lowest affinity anti-HER2 scFv, G98A, with a reported $K_D$ of 320 nM (ref. 35) to HER2 and a barely detectable FI signal at 100 nM HER2 (Extended Data Fig. 2c) as a starting point for affinity maturation by building six G98A CDR H3 libraries (Fig. 4a). On deep-screening and subsequent CDR sequencing, we detected $3 \times 10^5$ unique UMIs, coding for $2.4 \times 10^5$ unique CDR H3 sequences (of $6.2 \times 10^6$ possible CDRs; Fig. 4b). Despite sampling <5% of the potential diversity, principal component analysis (PCA) of the CDR H3 sequence space mapped to deep-screening data showed that function is highly localized to three fitness peaks in close proximity to each other, with the majority of mutations showing no detectable binding at the highest concentration tested (100 nM HER2; Fig. 4c).

Inspection of the three-highest-scoring clones (HER20003, HER20004 and HER20005, as judged by FI at 100 nM HER2) yielded

binding curves that closely match ML3-9 from the affinity panel with a known $K_D$ of 1.0 nM (Fig. 4d,e and Supplementary Fig. 11)[35], thus suggesting similar affinities but with dissimilar sets of mutations (Fig. 4d). These three scFvs were converted into Fabs, expressed in CHO cells[36], purified and binding kinetics measured by BLI, yielding $K_D$s of 2.8 nM for HER20003, 3.4 nM for HER20004 and 1.8 nM for HER20005, closely matching deep-screening observations (Fig. 4d,f and Supplementary Figs. 11 and 12). Thus, as observed previously, deep screening was successful in identifying antibodies with substantially increased affinity in a single experiment, in this case a 100-fold affinity maturation over the parental clone, G98A. However, our primary motivation for the above experiment had not been affinity maturation, but rather the generation of a large deep-screening dataset (HER2affmat) linking CDR H3 sequence (genotype) to HER2 binding affinity (phenotype) (comprising $2.4 \times 10^5$ genotype or phenotype pairs) as an input for ML and in silico generation of higher-affinity HER2 binders.

We selected a language model for this task because of the demonstrated capacity of language models for feature extraction and ability to 'learn' the underlying rules and hidden patterns within complex datasets in a self-supervised manner[37–39]. Numerous studies have shown the utility of large language models for the prediction of protein structure and function[40–47]. These models operate by first pre-training on a self-supervised task, such as filling in missing amino acids (AAs) or predicting the next AA in a protein sequence. Pre-training is typically accomplished using large protein sequence databases, such as UniProt, and randomly masking or mutating some percentage of AAs in each sequence, with the goal of learning the general underlying rules and global sequence patterns that give rise to functional proteins. Such pre-trained models may then be applied to more specific applications, such as protein structure and function prediction, through a second fine-tuning process. We hypothesized that a language model that has been pre-trained on a large antibody-sequence dataset, such as the Observable Antibody Space (OAS) dataset[48,49], would impart implicit representation of the rules that govern functional human antibody sequences that have passed the B cell maturation and quality-control processes, and that such a model might be able to leverage this representation to make accurate predictions of antibody-binding affinities when fine-tuned on deep-screening datasets.

To test this hypothesis, we built an 86-million-parameter Bidirectional Encoder Representations from Transformers (BERT) model[50] inspired by ProtTrans[46] that we termed BERT-DS (Extended Data Fig. 4). A BERT architecture was selected because—unlike a basic transformer model—it is capable of learning the underlying statistical representations of a sequence or language (words, sentences or AAs) in an unsupervised manner and then later solve a downstream task through a process called fine-tuning. We pre-trained BERT-DS to solve a masked language-modelling problem, where at each position in a protein sequence there is a 15% probability that the position will either be masked (80% probability), randomly mutated (10%) or have no change (10%). The BERT-DS model is tasked with predicting the ground truth of the masked or mutated AA. As a training dataset, we used $2 \times 10^7$ human heavy chain (VH) sequences from the OAS dataset[48,49] (Fig. 5a). Once BERT-DS had been pre-trained on the OAS dataset, we validated the model by challenging it to correctly predict the missing sequences of masked antibody VH domain framework and CDR regions. Pre-trained BERT-DS scored 97.66% accuracy in predicting missing framework and CDR residues on 100,000 sequences that were excluded from training.

Next, we sought to leverage BERT-DS to predict HER2 binding by formulating a classification problem, where predictions are binned into three categories (non-hit, low hit and high hit) using deep-screening FI values at a single condition. For maximum separation between the categories, we chose the 5 minute wash condition (first wash step after binding 100 nM HER2) from the kinetic dissociation measurements (Figs. 4e and 5a). We selected threshold FI values (see Methods) such that the parental clone G98A was roughly centred in the low-hit

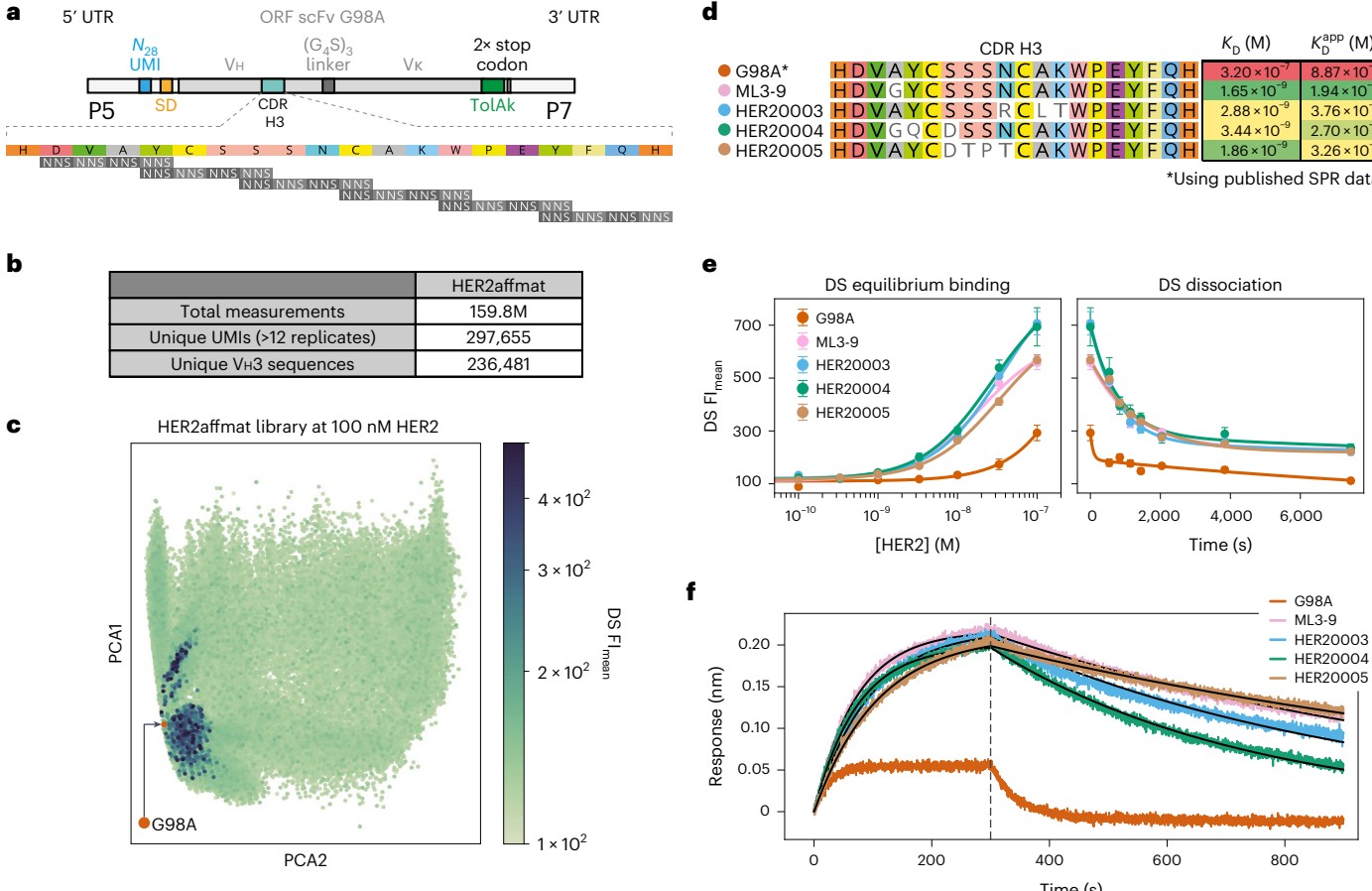

**Fig. 4 | Affinity maturation of the anti-HER2 scFv G98A. a**, Construct schematic of G98A, showing its CDR H3 sequence and a depiction of how the six scanning-window NNS sub-libraries were structured. **b**, Experiment statistics from the deep-screening component. **c**, PCA plot showing all 236,000 CDR H3 protein sequences projected into two dimensions and coloured by FI$_{mean}$ at 100 nM of HER2. A red dot shows the position of G98A wild type relative to the library. **d**, CDR H3 sequences of G98A, ML3-9 and three of the top-scoring clones identified by deep screening. As we were unable to obtain a 1:1 model fit to the BLI data of clone G98A at 20 nM of Fab, we opted to use the published surface plasmon resonance (SPR) $K_D$ value. Next to the sequences are binding $K_D$s identified via BLI, and the deep-screening-fitted equilibrium-binding $K_D^{app}$s. **e**, Deep-screening equilibrium-binding and kinetic dissociation curves showing G98A, ML3-9 and three of the top-scoring clones. Error bars are s.e.m. and $n \geq 12$ technical replicates of a given UMI. **f**, BLI kinetics of G98A, ML3-9 and three of the top-scoring clones at 20 nM of each clone in the Fab format on a HER2-loaded tip. The grey dashed line denotes the separation of the association (left) and dissociation (right) phases collected during BLI kinetics measurements.

category, and empirically adjusted the low-hit or high-hit threshold such that the high-hit category contained a sufficient number of sequences to enable training but would still comprise only the highest-affinity binders (Fig. 5a). This empirical adjustment improved model performance as too few high-hit sequences would reduce the capacity of the model to learn general rules of the high-hit category, whereas increasing the high-hit count too much would result in the model predicting sequences with a wider range of affinities than desired. With these thresholds set, the HER2affmat dataset yielded 232,693 non-hit, 1,284 low-hit and 111 high-hit V$_H$ sequences.

To perform classification with BERT-DS, we extended the model by attaching a classifier module to the last transformer block and used it to predict hit-category probabilities for a given input sequence (model architecture in Extended Data Fig. 4, details in Methods). We then fine-tuned the model on a train–test split of 90:10 on the HER2affmat dataset with early stopping to minimize overfitting. Our best BERT-DS model yielded $F_1$ scores (a measure of classifier accuracy and defined as the harmonic mean of precision and recall) of 0.993, 0.329 and 0.480 for the non-hit, low-hit, and high-hit categories, respectively, on the test set (Supplementary Tables 2 and 3). Although the $F_1$ scores for the low and high hits were less than ideal, they were dominated by their high

false positive rate, which is probably due to the challenge of defining the class boundaries across a continuous space of measurements. BERT-DS was able to accurately predict clones ML3-9, H3B1 and B1D2+A1 as high hits, although they were not present in the HER2affmat dataset.

Having established a fine-tuned BERT-DS model, we explored whether it could be used to generate anti-HER2 CDR H3 sequences with higher affinities than those observed in the HER2affmat dataset and how its performance compared with simple random mutagenesis. To this end, we took the three top-scoring clones (seeds) from the HER-2affmat dataset (HER20003, HER20004 and HER20005) and generated $1.98 \times 10^6$ mutant CDR H3 sequences in silico for each seed (Fig. 5b). Specifically, we generated all single, double, and triple mutants and up to $10^8$ fourth- and fifth-order mutants randomly. All $5.94 \times 10^8$ mutations were scored by BERT-DS before selections were made for a subsequent round of deep screening.

To compare the performance of BERT-DS with random mutagenesis, we devised a selection scheme where, for each seed sequence, a random mutation set was compiled from all 380 single mutants and up to 1,000 double, triple, and fourth- and fifth-order mutants each. This yielded a pool of 13,121 randomly mutated CDR H3 sequences (termed random/mut). Next, we assembled a pool of CDR H3 sequences with

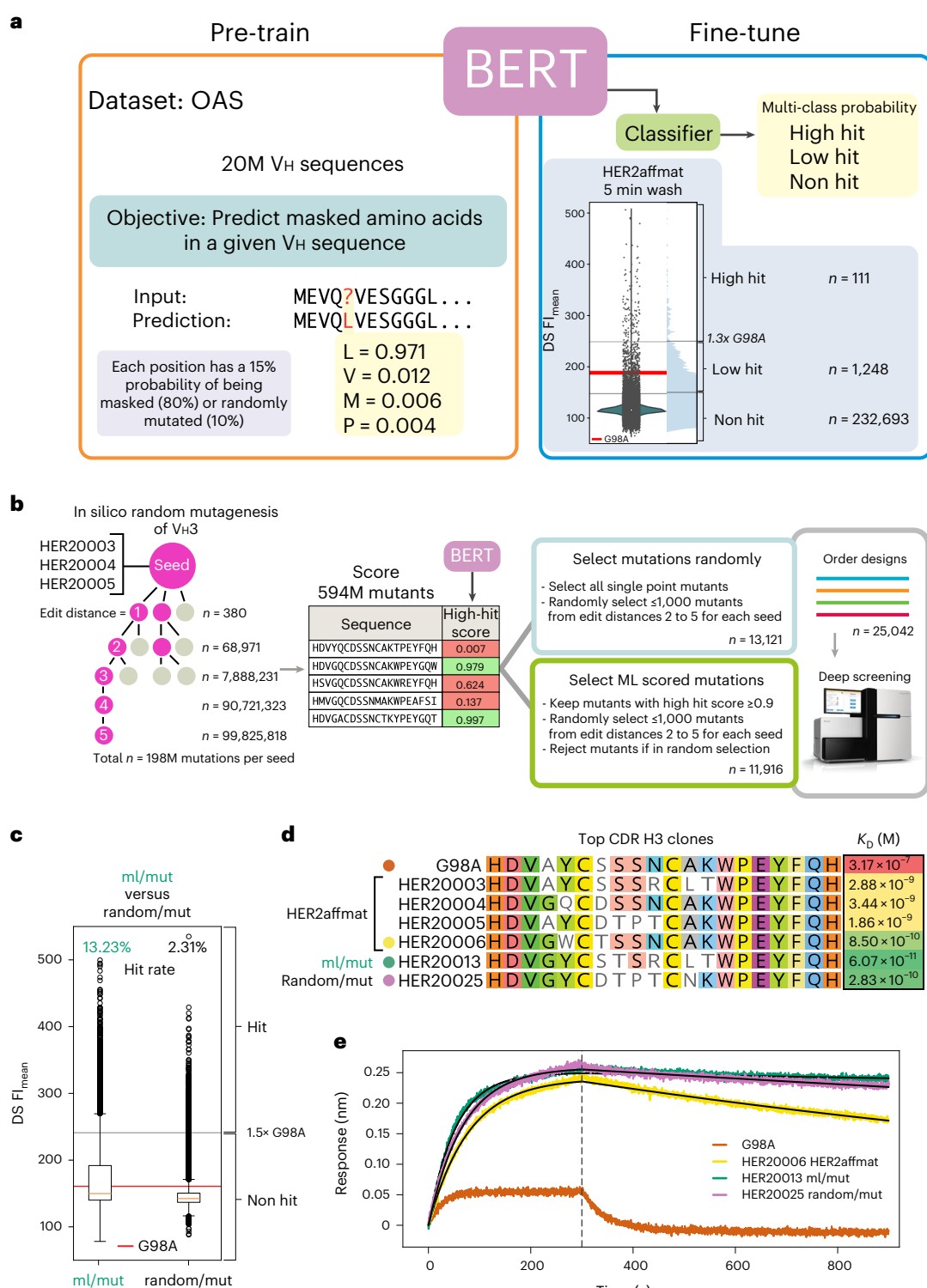

**Fig. 5 | ML-augmented antibody-affinity maturation. a**, An overview of the workflow used to pre-train the BERT-DS model on 20 million $V_H$ sequences from OAS on an MLM objective and to fine-tune the same model for classification of anti-HER2 binding. In the HER2affmat 5 min wash condition, a violin plot shows the data distribution in dark green, and an enlarged version of the same data is shown in light blue to better reveal the distribution. The red line shown in the violin plot indicates the $FI_{mean}$ value measured for clone G98A, with lighter grey lines indicating $FI_{mean}$ values 1.3× above and below. These lines were used to draw the hit thresholds for BERT-DS classification. **b**, The workflow used for in silico mutagenesis of three anti-HER2 seed sequences and selection of 13,121 random mutations and 11,916 ML-guided mutations before a second round of deep screening. **c**, Evaluation of the selected ML and random mutations at the 5 minute wash condition, which shows a substantial, 5-fold shift in the binding distribution of the ML-selected mutants (green) relative to making random mutations. In the box-and-whisker plots, the box extends from the lower to the upper quartile values with a red line to denote the median, and the whiskers extend to 1.5× the interquartile range. Outliers to the data are shown as a small open circle. **d**, CDR H3 sequences and BLI-derived $K_D$s of G98A, HER20003, HER20004, HER20005 HER20006, HER20013 and HER20025. **e**, BLI-derived binding kinetics of a HER2-loaded tip and the top-scoring clones from each library (as purified Fabs) at a concentration of 20 nM. The grey dashed line denotes the separation of the association (left) and dissociation (right) phases collected during BLI kinetics measurements.

exclusively ML-generated mutations by removing all sequences with a suboptimal high-hit score <0.9 and randomly selecting up to 1,000 double, triple, and fourth- and fifth-order mutants each, but excluding those that had already been selected in the random/mut set. This produced a pool of 11,916 CDR H3 sequences with ML-guided mutations (termed ml/mut; Fig. 5b). Finally, we included clones G98A, ML3-9, HER20003, HER20004 and HER20005 as internal benchmarks. This resulted in a total of 25,042 CDR H3 sequences that we ordered as a synthetic oligonucleotide pool (Supplementary Data 1) and subjected to deep screening for validation (with the same conditions as in the HER2affmat library).

In deep screening, we observed 24,968 (of 25,037) clones from the random and ML library (random/mut and ml/mut, 99.72% coverage), and 174,700 additional mutants due to errors in oligonucleotide synthesis and cloning (199,737 unique CDR H3 sequences in total). BERT-DS-selected CDR H3 sequences (ml/mut) showed a marked improvement in FIs compared with random mutagenesis (random/mut), not just in number of top FI signals but with a substantial overall shift in the global FI signal distribution towards high FI values (Fig. 5c). With reference to the previously established correlation between affinity and FI signals, this strongly suggested that the BERT-DS model had successfully distilled the salient features of high-affinity HER2 binding from the HER2affmat dataset and leveraged it to predict a large number of new, high-affinity HER2 binders.

As our aim was to leverage ML for the discovery of antibodies with higher affinities than the parental G98A, HER20003, HER20004 and HER20005 clones, we evaluated the deep-screening data as a binary classification problem, with G98A now being centred in the non-hit category and clones with intensities ≥1.5× above G98A classified as hits (Fig. 5c and Supplementary Fig. 14a,b). The resulting classification threshold showed an overall hit performance of 13.23% for the ml/mut clones versus 2.31% for the random/mut clones (Fig. 5c and Supplementary Table 4). Inspection of the number of hits per edit distance from the seeds showed that BERT-DS improved sequence space sampling between 2.6-fold and 32.6-fold over random mutagenesis, with a mean improvement over all edit distances of 5.7-fold and markedly improved performance at higher edit distances from the parent clones (Supplementary Table 4 and Supplementary Fig. 14a).

We selected 21 new anti-HER2 scFv clones (6 from the original HER2affmat library (HER20006–HER20011), 9 from the ml/mut set and 6 from the random/mut set) for conversion to Fabs for expression in CHO cells, purification, and characterization and affinity determination (Extended Data Fig. 3 and Supplementary Figs. 11–13). These clones were selected based on a variety of criteria, such as peak FI at all concentrations, equilibrium-binding and dissociation kinetics (Source data). Upon characterization, we observed the strongest correlation between BLI $K_D$ and deep-screening FI at 3.3 nM and at 10 nM HER2, with $r_s$ of −0.613 and a $P$ value of 0.002 (Extended Data Fig. 3d,e). We observed a weaker correlation between BLI $K_D$ and deep-screening-derived $K_D^{app}$ values, with $r_s$ of 0.563 and a $P$ value of 0.0027 (Extended Data Fig. 3f).

All the additionally selected clones from screening the HER2affmat library, including the three seeds (HER20003, HER20004 and HER20005) showed $K_D$ values from 850 pM to 5.25 nM and a general improvement in monomericity (93.5% for G98A compared with 94.4%–98.4% for the HER2affmat clones; see Source data for Extended Data Fig. 3 and Supplementary Figs. 12 and 13). Clones HER20006, HER20009 and HER20010 showed an ~300-fold improvement in affinity over G98A and had been selected for characterization due to their high FI values at low antigen concentration during the equilibrium-binding assay. Analysis of the ml/mut and random/mut library clones indicated a further improvement of affinity over the HER2affmat clones, with the top clone from the ML set (HER20013) showing a 5,220-fold improvement in affinity ($K_D$ = 60.7 pM) and another 4 clones (HER20015, HER20020, HER20021 and HER20022) from the ML

set showing a >1,000-fold improvement in affinity over G98A (Fig. 5d,e, Supplementary Fig. 12 and Supplementary Table 2). Although high-FI clones were >5-fold less frequent in the random set, we still managed to identify two clones (HER20024 and HER20025) with a >1,000-fold improvement in affinity over G98A, at 165 pM and 283 pM, respectively. In addition to affinity enhancement, we observed an overall improvement in monomericity for both the ML and random clones over G98A from 93.5% to 98.1% (Supplementary Fig. 13 and Source data). Taken together, these results show the exceptional effectiveness of combining deep screening with a natural-language ML model to discover high-affinity antibodies to a therapeutically relevant target at scale.

## Discussion

Building on work repurposing Illumina sequencing platforms for high-throughput screening[14–19], we have developed deep screening. Deep screening differs from previous strategies in the increased DNA and RNA cluster size (fragment length), the covalent attachment of the displayed RNA clusters to the flow cell and the potential for accessing much higher diversity, owing to capabilities of the HiSeq platform.

Deep screening enables the discovery of high-affinity antibodies directly from synthetic repertoires without any need for pre-selection or pre-enrichment, as we have shown for two human therapeutic drug targets (huIL-7 and HER2), with increases in affinity and inhibition potency of $10^3$- to $10^4$-fold in a single 3 day experiment. This was achieved even without accessing the full power of the HiSeq platform and using a relatively modest total repertoire diversity of ≤4 × $10^6$ (and even as low as 2 × $10^5$; Fig. 3) compared with classical bulk-selection techniques, such as mRNA, ribosome, yeast and phage display (with repertoire diversities typically ranging from $10^7$ to $10^{12}$). Therefore, even at relatively low library diversities, deep screening is capable of efficiently discovering rare and high-affinity clones typically present at <0.01% of repertoire diversity. We postulate that this is due to the high detection sensitivity achieved by interrogating antibody function arrayed in two dimensions and captured as digital data at multiple antigen concentrations during equilibrium binding and kinetic dissociation.

Such a digital readout of function for every library clone provides a great depth of information over a specific antibody–antigen interaction landscape and affords a global, granular view of the antigen-binding function across the whole library diversity. A digital signal readout also allows extraction of real-time antibody–antigen binding data with correction for confounding factors such as background noise, non-specific interactions, and artefacts such as dust, fibres, aggregates and flow-cell defects. Furthermore, readouts can be optimized for each specific antigen–antibody repertoire interaction, enabling exceptional and fine-grained control over the discovery process unavailable in bulk approaches. Combined with detailed sequence information for every cluster or antigen-binding signal and in-built repertoire redundancy (with ≥12 replicates), deep screening provides antigen-binding data for each of the library clones with statistical significance, greatly reducing the presence of false positives and other unwanted artefacts.

Even at the currently realized screening depth, deep screening may be sufficient to capture much of the diversity of a rodent or even a human antibody response. Of course, a screening depth of ≤4 × $10^6$ is a gross undersampling of the total potential VH, Vκ and Vλ diversity (estimated to be up to $10^{18}$) of the human repertoire[51]. However, although potentially all of the estimated $10^{11}$ circulating human B cells could display a different antibody[52], due to various biases the human immune system is estimated to mainly access only $10^6$–$10^7$ of actual antibody diversity at any point in time[53,54]; however, it should be noted that this number may be an underestimate as the above studies only sampled peripheral blood lymphocytes (which amount to only 2.5% of total blood lymphocytes) at the expense of less accessible lymphoid organs[55].

It would be desirable to access larger repertoires both for antibody discovery and for other biomolecular discovery projects because larger repertoires are generally thought to have a greater probability of containing high-affinity binders as they provide a more complete coverage of the shape space of possible epitopes[56,57]. Although deep screening has the potential for screening of repertoire diversities up to $3 \times 10^8$ (as shown here on a 2-lane HiSeq flow cell) and up to $2 \times 10^9$ on an 8-lane flow cell, we have not yet fully leveraged this potential diversity due to technical limitations, including library bottlenecking (to achieve 12-fold redundancy for data noise reduction). Future implementations of deep screening with improved display, antigen-binding and imaging protocols may enable screening at these or even higher depths on next-generation instruments.

A key aspect of the development of deep screening involved the optimization of protocols so that derived metrics, such as equilibrium-binding affinity ($K_D^{app}$) and peak fluorescence signal intensity (FI), correlate well with true antigen-binding affinities ($K_D$), as determined by state-of-the-art biophysical measurements on individually purified monovalent antibody Fabs. This enables not only reliable discovery of hits but also the use of deep screening to obtain internally consistent and reliable large datasets of global affinity and dissociation kinetics across an antibody repertoire. The latter is particularly important for ML-guided sampling of the antibody-sequence space. Indeed, we show that deep-screening datasets are of sufficiently high quality to be used as training data for ML. In particular, we developed a natural-language model based on the BERT architecture[50], and pre-trained it on the OAS database (to learn the general underlying rules and patterns of viable antibody sequences) and subsequently fine-tuned it using deep-screening data. The resulting trained model, termed BERT-DS, proved highly effective in generating high-affinity anti-HER2 binders (Fig. 5a), with an average improvement of more than 5-fold over random mutagenesis (Fig. 5c) and a top hit with a 5,200-fold increase in binding affinity for HER2 compared with the parental clone G98A (Fig. 5d,e).

Although pre-training natural-language models on naturally occurring protein sequences and predicting function is not a new concept, previous work had primarily focused on predicting improvements in proteins whose function is conserved in evolutionary history[43,45,58–61]. However, this approach is not readily applicable to antibodies, which are continuously generated de novo by the immune system with divergent antigen-binding functions. Thus, optimizing binding to a new target antigen is a more challenging task as the relevant information does not exist a priori within phylogeny. Although one could argue that the OAS database embodies general information on antigen binding, antigen-specific information is not available. In our implementation of BERT-DS, the antigen-specific sequence and functional scores are provided by deep-screening datasets, which can be readily collected.

Although our approach to sequence generation is relatively crude compared with state-of-the-art generative language or diffusion models[62–65], the combination of BERT-DS with subsequent deep screening of its predictions allowed for the discovery of high-affinity binders to a therapeutic drug target with an up to 32-fold improved success rate compared with random mutagenesis (depending on the number of mutations made; Supplementary Fig. 14a and Supplementary Table 4). Although many other ML approaches are available, we favoured a BERT model, which we considered to best capture the complexity of the genotype–phenotype association in antigen–antibody interactions. To critically test this assumption, we performed an ablation study (Methods, Supplementary Tables 5–14 and Extended Data Fig. 5) comparing the performance of BERT-DS (with and without pre-training on OAS) with a variety of other ML models. This showed that language models indeed performed best with our datasets, and indicates that BERT-DS with pre-training on OAS provides an increase in the prediction accuracy of antibody binding—still, a comparison

with BERT-DS trained several years apart suggests that follow-up studies are required to thoroughly evaluate whether pre-training on OAS results in major benefits.

An unexpected finding was that high-affinity antibodies isolated by deep screening typically also display desirable 'developability' features that are advantageous for antibody therapeutics, such as retention of affinity upon conversion to Fabs (or whole IgGs), a high degree of monomericity and high expression yields in CHO cells. We hypothesize that these features may arise due to a stringent pre-selection for desirable physicochemical properties by expression and folding during deep screening by the use of a minimal translation apparatus (devoid of chaperones) for 1 h at 37 °C, which predisposes hits to be fast and efficient folders. We also speculate that the crowded environment within each RNA cluster may mimic the intracellular environment and disfavour misfolded or incompletely folded proteins.

Although deep screening is currently implemented on the expiring HiSeq 2500, the approach is by no means restricted to this platform and should be extendable to related platforms such as those developed by Singular Genomics, Element Biosciences, Ultima Genomics and MGI, among others. Indeed, there are many technological aspects of the HiSeq platform that are suboptimal for deep-screening purposes, notably the imaging system, which is designed for thresholding fluorescence rather than for quantitative measurements. Although we currently perform sequencing, display, antigen binding and imaging on the same instrument, both internal and external imaging are possible, as shown for the MiSeq platform[19,66]. The use of external imaging devices could enable faster localized imaging across a wider range of colour channels and imaging modes. Faster imaging of large ($10^7$–$10^9$) $V_H \times V_L$ libraries might also bring direct screening of the naive rodent or human antibody repertoires within reach, with the potential to isolate antibody leads for a wide range of targets from a single deep-screening experiment.

In conclusion, deep screening expands the power of high-throughput phenotype screening into the realm of $>10^8$ simultaneous measurements. Together with methodological advances, this method allows for the display and direct screening of unselected antibody libraries with the discovery of high-picomolar binders in a 3 day experiment. Furthermore, deep screening generates large, internally consistent, genotype–phenotype correlation datasets that not only provide for efficient sampling of antibody-sequence and paratope space but also enable ML models to predict new sequences with improved affinities that are not present in the starting library. We anticipate many applications for deep screening, in particular the accelerated discovery and development of high-affinity antibodies for use in biotechnology and medicine, and as a multimodal tool for the exploration of the genotype–phenotype landscape of a wide range of biopolymers.

## Methods

### Construct design

To transcribe and translate sequenced DNA clusters on an Illumina flow cell, our DNA constructs contained the following elements: a P5 adaptor, followed by a 28 nt unique barcode, a 27 nt unstructured spacer (5p UNS v2), a ribosome binding site, start codon, protein coding region, TolAk short linker, 2× stop codons, a 27 nt unstructured spacer (3p UNS v2) and the P7 adaptor (Supplementary Table 1).

### Cluster generation and barcode sequencing

Libraries as subsequently described were clustered on an Illumina HiSeq 2500 using a paired-end rapid-run flow cell (PE-402-4002, HiSeq PE Rapid Cluster Kit v.2; Illumina) at 6 pM, which typically results in 200 million reads. Although these flow cells are perfectly capable of being clustered to yield upwards of 300 million reads, in the downstream RNA synthesis and ribosome display steps, we chose to hybridize a fluorescent Atto 647N oligonucleotide (R2_atto647N; Supplementary Table 1) to the P7 adaptor of each cluster to enable

normalization of the binding assay. At densities higher than 200 million reads, our HiSeq 2500 is unable to reliably focus and image the flow cell when all RNA clusters are labelled.

Clustering and sequencing were performed as a paired-end, single-read run with no indexing for 28 cycles on read 1, and 0 cycles on read 2, and executed using the HiSeq Control Software (HCS v.2.2.68; Illumina). The flow-cell and clustering reagents were sourced from the HiSeq PE Rapid Cluster Kit v.2 (PE-402-4002; Illumina) and sequencing reagents were sourced from the HiSeq Rapid SBS Kit v.2 (FC-402-4023; Illumina).

With 12-fold redundancy and the need to cluster the same library on two separate flow cells, there is a theoretical maximum yield of $2.5 \times 10^7$ UMIs. However, through the need to undercluster our 2-lane flow cells, various losses and amplification biases in library preparation, we typically yield $1 \times 10^5$ to $4 \times 10^6$ UMIs that satisfy the 12-fold redundancy requirement. Regardless, the benefits of direct observation of binding at scale allowed for the identification of high-affinity binders in smaller-diversity libraries than one would typically expect.

### RNA synthesis

Following sequencing, we closed HCS and launched the HiSeq engineering software (Archimedes Test Software v.3.8.317.0; Illumina), initialized the instrument, homed the stage, set the chemistry module run mode to 'RapidRun' and set the flow cell temperature to 20 °C. We then pumped 120 µl of Illumina's Universal Scan Mix into the flow cell before automatic tilting, aligning and imaging the flow cell using the 'Bruno Scan' module. We did this specifically by setting the surface to 'dual lane', the scan velocity to 2.0 mm s$^{-1}$ and the swath to 'dual swath'. The flow-cell images were saved and enabled us to measure offsets and chromatic aberration distortions between the different optical paths of the instrument.

We then denatured the sequencing product with a Fast Denaturation Reagent (FDR; Illumina) wash at 65 °C, followed by running the 'End Deblock' protocol as found in the HiSeq 2500 rapid-run recipe files generated by HCS, which uses the reagents Cleavage Reagent Mix and Cleavage Wash Mix to remove the remaining dye-terminated nucleotides that are still present on the flow-cell surface. With a single-stranded DNA template present on the flow cell, we then needed to 'deprotect' or remove the 3'-phosphate group from the P5 primer. This was done using the Fast Resynthesis Mix and the deprotection protocol.

With a free 3'-hydroxyl group on the P5 grafted primer, we repurposed the paired-end turnaround process and performed a cycled RNA primer extension using TGK polymerase. Here TGK takes a DNA primer (grafted P5) annealed to a DNA template (cluster strands) and the primer is extended with ribonucleotides (NTPs). This was done by heating the flow cell to 55 °C and performing 12 cycles of injecting FDR, annealing and extension with the TGK Amplification Mix (TAM; 625 µM NTPs, 10 nM TGK, 18 U ml$^{-1}$ Superase In (AM2696; Thermo), 2 M betaine, 20 mM Tris, 10 mM ammonium sulfate, 6 mM MgSO$_4$, 0.1% Triton-X and 1.3% DMSO, pH 8.8); each extension step had an incubation time of 1,800 seconds.

After 12 cycles of RNA extension, we observed that for long templates (>900 nt), TGK is unable to completely synthesize the strand. We believe this to be due to a build up of torque in the DNA–RNA duplex that is covalently attached to the surface via the respective 5' ends. To relieve the torque, we annealed an oligonucleotide over the 8-oxoguanine site on the grafted P7 primer and performed 2 cycles of cleavage with Illumina's FLM2 (Fast Linearisation Mix 2) reagent and extension (with TAM) at 37 °C for 30 minutes and 55 °C for 1 hour, respectively (P7'_surface_hyb; Supplementary Table 1).

Following DNA cleavage and final extensions, we denatured the DNA–RNA duplex and washed away the DNA template Illumina's FDR mix. With clusters of single-stranded RNA present on the flow cell, 100 nM of R2_atto647N was annealed to the P7 adaptor at the 3' end of each molecule of RNA.

### Ribosome display on an Illumina flow cell

Ribosome display was performed using a custom PURExpress kit from New England Biolabs (NEB) that lacks release factors 1, 2 and 3, and T7 RNA polymerase. Specifically, we prepared a 200 µl master mix containing 80 µl of solution A, 60 µl of solution B, 4 µl of disulfide enhancers 1 and 2 (E6820S; NEB) (if required) and 4 µl of Superase In (AM2696; Thermo Fisher). We then injected the master mix into each lane of the flow cell, being careful to avoid the introduction of bubbles, before incubating the flow cell at 37 °C for 60 minutes. Once the incubation period was complete, we cooled the flow cell down to 20 °C, before washing and stabilizing the ribosomes with ribosome display buffer (50 mM Tris(hydroxymethyl)aminomethane acetate, 150 mM NaCl, 50 mM magnesium acetate, 0.1% Tween 20 and 1 U ml$^{-1}$ of Superase In (AM2696; Thermo Fisher), pH 7.5).

With the ribosomes stabilized by the display buffer, we block the flow cell with binding buffer (ribosome display buffer with 0.1% BSA; A9647; Sigma-Aldrich). After flow-cell blocking, we image the surface to determine a baseline for background fluorescence.

### Sequencing of CDRs via internal sequencing primers

Following a successful deep-screening display experiment, we set-up a second sequencing experiment on a fresh flow cell using the same library for resolving the CDR sequences with internal sequencing primers. CDR sequencing experiments were performed in HCS with a custom recipe that initially sequenced the $N_{28}$ UMI with Illumina's Read 1 Sequencing Primer for 28 cycles, followed by denaturation of the sequencing product with FDR at 65 °C, annealing of an appropriate internal sequencing primer and sequencing enough cycles to cover the region of variability. All internal sequencing primers used in this work were ordered from IDT, HPLC purified and resuspended in IDTE at 100 µM.

### Image processing

A technical challenge lay in the nature of the HiSeq instrument, which is not designed for quantitative measurement; rather its epi-fluorescence line-scanning imaging system is designed to threshold fluorescence signals between four colour channels to determine base calls during sequencing. We solved this challenge for quantitative measurement of binding interactions by preparing libraries with 12-fold redundancy of each UMI; that is, each UMI considered during analysis was present at least 12 times at different locations on the flow cell.

Furthermore, the HiSeq is an epi-fluorescence line-scanning microscope with 532 nm and 660 nm lasers that requires a substantial amount of illuminated signal in the 660 nm channel (as expected during a sequencing run) to first locate the flow-cell surfaces and then maintain focus during a scan. This imaging mode is poorly suited for the screening of binding interactions, where clusters showing a high signal are rare and do not provide sufficient signal for focusing. We solved this problem by labelling all RNA clusters through hybridization of a fluorescently labelled DNA oligonucleotide to the 3' end, enabling focused imaging of the whole flow cell even with only sporadic or no cluster signal in the 532 nm channel (which we use to detect protein binding). In addition, this approach provided us with a diagnostic for RNA-synthesis efficiency or cluster size. The ability to conduct all steps (comprising sequencing, RNA and protein synthesis, and imaging) within the same instrument streamlines the experimental pipeline and avoids challenges with image alignment.

A single scan of a 2-lane rapid-run flow cell generates $8 \times 2,048 \times 160,000$ pixel 16 bit TIFF images in 4 colour channels, for a total of 32 images. The HiSeq 2500 uses a 532 nm and 660 nm laser with a set of emission filters that path out to 4× time-delayed integration-line-scanning CCD detectors. We can detect signal from Alexa Fluor 647 or Atto 647 on the 'A' and 'C' channels, and Alexa Fluor 532 (AF532) on the 'G' and 'T' channels, with the highest signal-to-noise ratio observed on the C and T channels with these dyes. As such, we only perform analysis using the C and T colour channels.

Our image-processing pipeline operates by breaking up each of the $2,048 \times 160,000$ pixel images into 16 tiles that are processed independently in parallel. For a given tile image, we first perform a non-uniform illumination correction by applying a morphological opening with a disk-shaped structuring element using a radius of 25 pixels before subtracting the morphological opening from the tile image. We then detect the centroids of any clusters present in the tile image using a peak local maximum function that operates by initially performing a morphological dilation of the tile image with a $3 \times 3$ pixel square kernel. The algorithm then moves through each pixel of the tile image and checks if the pixel is equal to the value of the same dilated pixel, and whether that pixel intensity is above a set threshold. If a given pixel meets these conditions, it is deemed to be a centroid, and is added to the centroid map. In this case, we are using a pixel intensity threshold of 600 (this value was manually tuned for our instrument). This method for cluster detection is simple, fast to compute and generally good enough.

Using the detected cluster coordinates on the C and T images, we align these against the known sequencing coordinates using a discrete-Fourier-transform phase correlation function from the OpenCV package. As there are some slight variations in the repeatability of the microscope stage and optical distortion within the HiSeq, we perform a refined alignment by subdividing the tile image further into $128 \times 128$ pixel non-overlapping subimages and saving the refined offsets to an offset map.

Using the refined offset map, we quantify the intensity of every known cluster from the sequencing data by extracting a $9 \times 9$ pixel subimage centred on the offset-corrected cluster coordinates. We then perform an element-wise multiplication of the $9 \times 9$ pixel subimage with a $9 \times 9$ pixel array constructed from a two-dimensional (2D) Gaussian point spread function (PSF) with a sigma ($\sigma$; width of the Gaussian bell) of 0.5. We use the following equation to describe the 2D Gaussian PSF:

$$\mathrm{PSF} = 1\mathrm{e}^{\left(-\left(\frac{(cx-x)^2}{2\sigma^2} + \frac{(cy-y)^2}{2\sigma^2}\right)\right)}$$

Here, $cx$ and $cy$ is the centre of the Gaussian peak, $x$ and $y$ are the respective 2D coordinates. The sum of pixel values after the element-wise multiplication is what we define to be the cluster intensity. The image-processing pipeline reports cluster intensities for every sequenced cluster on the C and T channels from every scan of the flow cell and saves this to disc or inserts it into a database.

### Data analysis

Data analysis starts by grouping all cluster data by their common $N_{28}$ UMI. If there are at least 12 replicates, where a cluster has not been rejected for falling outside of the imaging area, the UMI is retained. Next, we group the UMI and binding data with the UMI and CDR sequencing data, where there exist at least three CDR reads per UMI. Following the grouping, CDR reads are consensus error corrected (and the UMI is dropped if there is no consensus) before performing median absolute deviation outlier rejection and calculating the mean, median, s.d. and s.e.m. for each UMI on both the T (532 nm; protein) and C (660 nm; RNA) colour channels.

Flow-cell-based equilibrium-binding curves ($K_D^{app}$) are fit using the following equation to the mean integrated intensities of a given UMI via least squares, as implemented in the curve_fit function from the python package SciPy:

$$R = \frac{F_{max}}{1 + \left(\frac{K_D^{app}}{x}\right)} + F_{min}$$

where $F_{max}$ is the maximum intensity observed, $F_{min}$ is the minimum intensity observed, $K_D^{app}$ is the equilibrium-binding constant that we wish to fit and $x$ is the concentration of a given measurement.

Flow-cell-based kinetic dissociation curves are fit using the following biphasic dissociation equation via least squares, as implemented in the curve_fit function from the python package SciPy:

$$R = R_1 \mathrm{e}^{(-k_{d1}(t-t_0))} + (R_0 - R_1)\,\mathrm{e}^{(-k_{d2}(t-t_0))}$$

where $R_0$ is the intensity observed at the start of dissociation, $R_1$ is a floating parameter for the initial intensity for component 1, $t$ is time in seconds, $t_0$ is the start time for the dissociation and $k_{di}$ is the dissociation rate constant for component $i$.

We chose a biphasic dissociation equation, due to the complex dissociation kinetics observed within the flow-cell environment. To elaborate, the flow cell is a heterogenous environment where clusters containing different antibody clones compete against each other for binding and rebinding during the wash conditions. In our evaluations, we found that a biphasic dissociation model best represents the kinetics observed. Future studies should examine this phenomenon in more detail.

### PCA

PCA plots were generated by a compressed one-hot vector encoding of all sequences identified from a given library and computing the first two principal components using the PCA.fit function from the scikit-learn python library. The first two principal components were then plotted as a 2D scatter plot using matplotlib, and points were coloured based on their $FI_{mean}$ values at the condition shown in the respective plots.

In more detail, our one-hot vector encoding scheme encodes each AA as a binary 1D vector that is 5 long. We chose this encoding scheme as it can fully capture all AAs, including stop codon and unknowns, while minimizing sparseness in the representation.

For example, alanine is encoded as [0, 0, 0, 0, 0] and glycine is encoded as [0, 0, 1, 0, 1].

To encode a full sequence, each AA encoding is appended to an array and finally flattened to a 1D vector.

### Nanobody yeast surface display selections

The nanobody yeast display library was acquired from the Kruse laboratory as a frozen stock of $>2.5 \times 10^9$ cells (EF0014-FP; Kerafast)[1]. The library aliquots were initially thawed at 30 °C, before being recovered in 1 l of 'Yglc4.5 −Trp' (3.8 g $l^{-1}$ −Trp yeast dropout media supplement (Y1876; Merck), 6.7 g $l^{-1}$ yeast nitrogen base (Y0626; Merck) and 10 ml $l^{-1}$ Penicillin-Streptomycin (P4333; Merck)), shaking at 230 rpm, 30 °C, overnight. The recovered culture was then expanded to 3 l of media and allowed to grow to a stationary phase (OD$_{600}$ of 20) over 48 hours. The culture was centrifuged at 3,500$g$ for 5 minutes and resuspended in fresh Yglc4.5 −Trp supplemented with 10% DMSO, such that the final density is $10^{10}$ cells per ml before making 2 ml aliquots and freezing at −80 °C.

To prepare the naive library for the first round of selection, one aliquot was thawed at 30 °C and used to inoculate 1 l of Yglc4.5 −Trp supplemented with 2% galactose. The culture was then grown for 72 hours at 24 °C. Expression was confirmed by flow cytometry with a FITC-labelled anti-HA antibody (GG8-1F3.3.1; Miltenyi Biotech) before the first round of selection. Cells representing over 10-fold the library diversity were initially deselected against SA microbeads (Miltenyi Biotech) for 1 hour at 4 °C in PBS−T−BSA (0.1% Tween 20 and 0.1% BSA) before being separated from the beads on a Miltenyi MACS magnet. Deselected cells were then incubated in the presence of 500 nM HEL−biotin (GTX82960-pro; GeneTex) for 1 hour at 4 °C. SA beads were added and incubated further for 15 minutes before selection and washing on a Miltenyi MACS magnet. Beads and the bound cells were eluted, pelleted and resuspended in 1 l of Yglc4.5 −Trp supplemented with 2% galactose before growth for 72 hours at 24 °C. Round 2 was conducted similarly to round 1, with the absence

of a deselection step and reduction to 300 nM HEL–biotin before adding SA microbeads, panning on a MACS column, and washing and recovering the cells.

After round 2, the recovered cells were split in half by volume to conduct a round 3 via MACS and FACS with the respective splits. Round 3 MACS was conducted as per round 2 with a further reduction to 200 nM HEL–biotin, followed by recovery, collection of cells by centrifugation and miniprep of the plasmid DNA (D2004; Zymo Research). Before collecting cells, 100 μl of cells was serially diluted and plated on YPD agar plates to enable picking of 96 colonies for colony PCR and Sanger sequencing. Round 3 FACS was conducted by incubating cells with 200 nM HEL–biotin for 1 hour at 4 °C, pelleting and resuspending cells in fresh PBS–T-BSA, combining with 100 μg of NeutrAvidin-PE (A2660; Thermo Fisher Scientific), and performing a 1:1000 dilution of the anti-HA–FITC antibody for 15 minutes before sorting on a Synergy 3 cell sorter (Sony Biotechnology) and gating for dual-labelled (FITC/PE) events, yielding 50,135 cells. Sorted cells were recovered and miniprepped as per round 3 MACS. A diagram of this selection scheme can be found in Fig. 2a.

### Nanobody library preparation and deep screening

Minipreps for round 3 MACS and FACS were PCR amplified (Q5 polymerase; M0492; NEB) for 20 cycles using primers that anneal with the amino-terminal framework region and carboxy-terminal HA tag, and introduce a 20 nt overhang at the 5′ end of each primer that contain homology with the 5′ flow-cell adaptor (RBS + ATG; KF_olap.fwd; Supplementary Table 1) and the 3′ flow-cell adaptor (TolAk linker; KF_olap. rev; Supplementary Table 1).

The nanobody library, now containing homology with the adaptors, was run on a 1% agarose gel and a band of approximately 449 bp was gel extracted (approximate because the library contains variably sized CDR loops), purified and quantified by NanoDrop. The library is subsequently assembled into the deep-screening display construct via Gibson assembly using 0.2 pmol of the 5′ adaptor, the nanobody library fragment, 3′ adaptor and the HiFi DNA Assembly Master Mix (E2621; NEB) and incubated at 50 °C for 30 minutes. The library is then bottlenecked by taking 300 amol of material from the Gibson assembly reaction (assuming 100% assembly efficiency) and PCR amplifying for 25 cycles with Q5 polymerase and the outnest P5 and P7 primers (Supplementary Table 1).

The PCR product was run on a 1% agarose gel and a roughly 800 bp band was gel extracted, purified and quantified initially by NanoDrop and subsequently by quantitative PCR (qPCR; NEBNext Library Quant Kit, E7630; NEB).

The quantified library was diluted to 2 nM before being denatured (10 μl of library was mixed with 10 μl of 100 mM NaOH and incubated at room temperature (RT) for 5 minutes) and rapidly diluted to 20 pM in HT1 buffer provided by the rapid PE flow-cell clustering kit (PE-402-4002; Illumina). We diluted the library to a final concentration of 6 pM before loading into the template slot on the HiSeq 2500 and setting up a deep-screening experiment as described above and below.

Following acquisition of the baseline flow-cell images, we performed an equilibrium-binding assay at successive and increasing concentrations of HEL–biotin. Specifically, each condition involves an injection of 120 μl of HEL–biotin (GTX82960-pro; GeneTex) that had been pre-complexed with AF532–SA (S11224; Thermo Fisher) at a 1:1 ratio in display buffer at 20 °C, an incubation of 45 minutes at 20 °C, and a 200 μl wash of display buffer, followed by complete imaging of the flow cell. This was performed for 1 nM, 10 nM, 100 nM and 300 nM HEL with 1:1 amounts of AF532–SA. Following the highest concentration of HEL, we proceeded to collect measurements for a kinetic dissociation rate. This was accomplished by pumping display buffer over the flow cell and imaging at 5 minutes, 10 minutes, 15 minutes, 20 minutes, 30 minutes, 60 minutes and 120 minutes. Raw images were then processed as described above.

### Nanobody expression and periplasmic extraction

Nanobody hits (as defined in Extended Data Fig. 1) were computationally composed, assuming no mutations were present outside of the sequenced CDR regions, which contains 3 nt before and after the actual variability. Composed hits were then codon optimized and ordered as a gBlock from IDT (Integrated DNA Technologies, Inc) before being cloned via FX (fragment exchange) cloning into the *E. coli* periplasmic expression vector pSBinit, a gift from Markus Seeger (Addgene plasmid number 110100) (refs. 2,3). Single colonies were picked, and correct clones were validated by Sanger sequencing. Following validation, single colonies were grown overnight in a 24-deep-well plate with 5 ml of LB and 25 μg ml$^{-1}$ chloramphenicol at 37 °C before being subcultured at 1:100 into 5 ml of TB (with chloramphenicol). Cultures were grown at 37 °C and induced roughly at an OD$_{600}$ of 0.6–0.9 with 0.05% w/v L-arabinose. Cultures were grown for another 3.5 hours before being collected by centrifugation at 2,500$g$ for 20 minutes at 4 °C and supernatant discarded. Pellets were resuspended (1/20 of the original culture volume) in 250 μl TES buffer (50 mM Tris–HCl, pH 7.2, 0.1 mM EDTA and 20% sucrose) and incubated on ice for 60 minutes to perform a periplasmic extraction. The supernatant was then collected by centrifugation at 4,000$g$ for 30 minutes at 4 °C and protein yield was quantified by SDS–PAGE. All clones were normalized to a concentration of 500 nM in SuperBlock PBS (37515; Thermo Fisher Scientific) before BLI kinetics measurements.

### Nanobody kinetics measurements

Periplasm-extracted nanobodies that had been normalized to 500 nM in SuperBlock PBS were further diluted to 50 nM. BLI kinetics were performed on an Octet Red384 (Sartorius) with reference subtraction performed for each nanobody clone using a non-loaded SA tip (18-5136; Sartorius). Kinetics were measured using the following steps: (1) sensor check for 30 seconds, (2) loading of HEL–biotin at 25 μg ml$^{-1}$ for 400 seconds, (3) baseline measurement for 240 seconds, (4) association kinetics at 50 nM of each nanobody for either 400 seconds or 500 seconds, and (5) dissociation kinetics for 600 seconds. In all stages, SuperBlock PBS was used as the buffer.

### BLI data fitting

BLI kinetics data were collected on an Octet Red384 instrument as described in the previous and subsequent kinetics measurements sections. In all cases, SA tips (18-5136; Sartorius) were loaded with biotinylated target antigen and washed to a baseline signal before binding at a fixed concentration of each VHH or Fab clone. After collection of on rate kinetics, tips were dipped in fresh buffer to measure off rate kinetics. Measurement data for each clone were referenced against SA-only tips to remove non-specific binding to SA.

A 1:1 binding model was fit to all data via least squares using SciPy. Association rates were fit to the following equation:

$$R_{assoc} = R_{max}\left(\frac{1}{1 + \frac{K_d}{K_a C}}\right)\left(1 - e^{(-K_a C K_d)t}\right)$$

where $R_{max}$ is the peak response, $K_d$ is the dissociation rate to be estimated, $K_a$ is the association rate to be determined, $C$ is the concentration of the Fab in molar and $t$ is time in seconds.

Dissociation rates were fit to the following equation:

$$R_{dissoc} = Y_0 e^{-K_d(t-t_0)}$$

where $Y_0$ is equal to $R_{assoc}$ at the end of the association phase, $K_d$ is the dissociation rate to be determined, $t$ is the current time in seconds and $t_0$ is the time at the start of the dissociation phase.

$K_D$ values are calculated as:

$$K_D = \frac{K_d}{K_a}$$

## IL-7 library preparation and deep screening

The unselected IL-7 Vκ light-chain CDR L1 and L3 scFv library was prepared and provided to us by AstraZeneca in the pCANTAB6 plasmid. The scFv library was extracted by 20 cycles of PCR using Q5 polymerase and primers that provide 25 nt of homology with the 5′ and 3′ display adaptors. The PCR product was run on a 1% agarose gel, and a roughly 778 bp band was gel extracted and purified. Similar to the nanobody library assembly, 0.2 pmol of the 5′ adaptor, the scFv library fragment, 3′ adaptor and the HiFi DNA Assembly Master Mix (E2621; NEB) were combined and incubated at 50 °C for 30 minutes. The library is then bottlenecked by taking 500 amol of material from the Gibson assembly reaction (assuming 100% efficiency) and PCR amplifying for 25 cycles with Q5 polymerase and the outnest P5 and P7 primers. The PCR product was run on a 1% agarose gel and a 1.2 kb band was gel extracted, purified, and quantified initially by NanoDrop and subsequently by qPCR (NEBNext Library Quant Kit, E7630; NEB).

The quantified library was diluted to 2 nM before being denatured (10 μl of library is mixed with 10 μl of 100 mM NaOH and incubated at RT for 5 minutes) and rapidly diluted to 20 pM in HT1 buffer provided by the rapid PE flow-cell clustering kit (PE-402-4002; Illumina). We diluted the library to a concentration of 6 pM before loading on the HiSeq 2500 and setting up a deep-screening experiment as described above.

Following acquisition of the baseline flow-cell images, we performed an equilibrium-binding assay at successive and increasing concentrations of huIL-7–biotin pre-complexed with AF532–SA (S11224; Thermo Fisher) in a 1:1 ratio (100 pM, 333 pM and 1 nM). In this experiment, we observed substantial aggregation of huIL-7 on the flow-cell surface that prohibited imaging past a concentration of 1 nM huIL-7; as such, no kinetic dissociation measurement was collected. Images were processed and CDR sequences resolved as described above, which we used to identify putative hits.

## Anti-IL-7 and anti-HER2 Fab expression and purification

The top-19 putative anti-IL-7 hits (and IL70001) and all 26 anti-HER2 hits (including G98A and ML3-9) were converted from scFv to Fab format, with the VH and VL variables being synthesized separately and cloned into mammalian expression vectors pEU10.1 and pEU4.4, respectively. Vectors were transiently transfected into CHO cells using PEI and a proprietary medium. Expressed Fabs were purified by loading the cleared culture supernatant onto a CaptureSelect CH1-XL column (Life Technologies; Thermo Fisher), running in DPBS, eluting with 25 mM acetate (pH 3.6), and buffer exchanging into DPBS (pH 7.4) using PD-10 desalting columns (Cytiva). The concentration was determined spectrophotometrically using an extinction coefficient based on the AA sequence. The protein purity was verified by SDS–PAGE and the verification of correct molecular weight was achieved by LC-MS analysis[4]. Analytical high-performance size exclusion chromatography was performed post-purification by loading 70 μl of each protein onto a TSKgel G3000SWXL, 5 μm, 7.8 mm × 300 mm column, using a flow rate of 1 ml min⁻¹ and 0.1 M sodium phosphate dibasic anhydrous with 0.1 M sodium sulfate, pH 6.8, as the running buffer. A gel filtration standard (151-1901; Bio-Rad) was also run for comparative purposes.

## IL-7 kinetics measurements

Kinetics of binding for the top-19 hits and IL70001 was measured using Octet BLI and SA-coated tips (18-5136; Sartorius). In all cases, the buffer used was DPBS (14190-169; Gibco) with 0.1% BSA and 0.02% Tween 20. Purified Fabs were diluted to a final concentration of 50 nM. Kinetics were measured using the following steps: (1) sensor check for 60 seconds, (2) loading of huIL-7–biotin at 5 μg ml⁻¹ for 30 seconds,

(3) baseline measurement for 60 seconds, (4) association kinetics at 50 nM of each Fab for 300 seconds, and (5) dissociation kinetics for 600 seconds.

## TF-1 STAT5 IL-7Rα and IL-7Rγ cell-based reporter assay

Two vials containing 1 ml of $10^7$ per ml TF-1 STAT5 IL-7α and IL-7γ luciferase cG3 cells were removed from liquid nitrogen, defrosted, transferred into 50 ml Falcon tubes (2 vials per tube) containing 40 ml of complete medium, and centrifuged for 5 minutes at 1,200 rpm. The supernatant was aspirated, and cell pellets were resuspended in 40 ml RPMI (11875093; Thermo Fisher) with 10% FBS and 1% sodium pyruvate before centrifugation for another 5 minutes at 1,200 rpm before aspirating the supernatant as before. Cells were finally resuspended in 40 ml RPMI with 10% FBS and 1% sodium pyruvate, placed in a T175 flask and incubated for 24 hours at 37 °C in an atmosphere of 5% $CO_2$.

huIL-7 (CHO expressed) was made up to 0.12 nM in RPMI with 10% FCS and sodium pyruvate, which was then diluted 1:100 to a final volume of 20 ml for addition to a 384-well plate. Purified Fabs were added undiluted to a 384-well plate, and an 11-point, 3-fold duplicate serial dilution was performed using a Bravo liquid handling platform into complete RPMI. Cells were removed following the 24 hour incubation and pelleted by centrifugation at 1,200 rpm for 5 minutes and resuspended in 10 ml of RPMI with 10% FCS and 1% sodium pyruvate. Cells were counted and diluted in complete RPMI to give a concentration of 10,000 cells per 20 μl. Cells (20 μl) were then added to 3× 384-well clear assay plates. A 10 μl volume of the titrated Fabs was added to the cells, followed by 10 μl of 120 pM huIL-7. The plates were then placed in a tissue culture incubator for 6 hours at 37 °C in an atmosphere of 5% $CO_2$. Steady-Glo reagent (100 ml; E2520; Promega) was defrosted before use and 40 μl was added to each well of the 384-well plates. The plates were sealed and incubated for 10 minutes in a plate shaker before measurement. Luminescence readings were measured using an EnVision plate reader with a 1 second pulse time. Each Fab was measured in duplicate.

Data were exported and processed, and mean data were fitted using least squares to a log inhibitor response curve defined by the following equation:

$$Y = \text{bottom} + \frac{(\text{top} - \text{bottom})}{\left(1 + 10^{((\log IC_{50} - X)\text{HillSlope})}\right)}$$

where $Y$ is the response, bottom is the response at the minimum of the sigmoid curve, top is the response at the maximum of the sigmoid curve, $\log IC_{50}$ is the log concentration of the inhibitor that gives a response halfway between the top and the bottom, and HillSlope describes the steepness of the curve. $X$ is the experimental concentration of the inhibitor.

## Deep screening of the anti-HER2 affinity panel

The anti-HER2 scFv affinity panel plus trastuzumab[5] protein sequences were back translated, codon optimized and composed into the deep-screening display construct with a known 28 nt UMI. DNA constructs were ordered as gBlocks from IDT and clustered on a rapid PE flow cell at 1% per construct, with the remaining clusters on the flow cell comprising PhiX control (FC-110-3001; Illumina). The flow cell was sequenced for 28 cycles and deep-screening display was conducted as described above.

Following successful display, we performed an equilibrium-binding assay using biotinylated human HER2 (HE2-H822R, 25 μg; Acro Biosystems) and AF532–SA (S11224; Thermo Fisher). In this instance, a binding assay cycle was conducted by injecting 120 μl of HER2–biotin, incubating for 45 minutes at 20 °C, washing with 200 μl of display buffer, injecting 120 μl of 100 nM AF532–SA, incubating for 10 minutes at 20 °C before washing with 200 μl of display buffer and imaging. The equilibrium-binding assay was performed at 100 pM, 333 pM, 1 nM,

3.33 nM, 10 nM, 33.3 nM and 100 nM HER2–biotin before initiating a kinetic dissociation assay. The dissociation assay was performed by pumping wash buffer over the flow cell and imaging at 5 minutes, 10 minutes, 20 minutes, 60 minutes, 240 minutes and 420 minutes. Data collected from this experiment were processed as described above, and aggregate statistics were calculated through grouping by the known UMIs.

## Anti-HER2 scFv affinity maturation library preparation and deep screening

We built a CDR VH3 affinity maturation library with G98A as the parental starting clone. This was accomplished by topoisomerase-based cloning (450245; Thermo Fisher) the G98A gBlock from the previous section into TOP10 Chemically Competent Cells (C404010; Thermo Fisher), picking 6 colonies, growing these overnight in 5 ml TB with 50 µg ml⁻¹ kanamycin and miniprepping 2 ml of culture. Plasmids were sent for Sanger sequencing using M13 forward and reverse primers; one of the correct colonies were taken forward for subsequent processing. As we wanted to build a VH3 affinity maturation library, we first needed to extract the regions upstream and downstream of VH3. We did this by PCR amplification of the plasmid DNA as two reactions for 25 cycles using Q5 polymerase with primer set 1 (G98A_olap.fwd and G98A_5p_VH3.rev; Supplementary Table 1) and primer set 2 (G98A_3p_VH3.fwd and G98A_olap.rev; Supplementary Table 1). Both PCRs were subsequently treated with DpnI (R0176L; NEB) for 1 hour at 37 °C before being purified with a PCR clean-up kit (T1030S; NEB). This process yielded the upstream and downstream fragments of the G98A clone with homology to the deep-screening display construct while removing contaminating wild-type plasmid DNA.

We next assembled the HER2 affinity maturation library by 20 cycles of PCR using Q5 polymerase, the upstream and downstream fragments of G98A, an equimolar amount of VH3 NNS oligonucleotides that produce a scanning window of 4 NNS codons across the CDR VH3, and the G98A overlap forward and reverse primers (Supplementary Table 1). This product is then column purified using a PCR clean-up kit (T1030S; NEB). We next append the deep-screening 5′ and 3′ adaptors using the Gibson assembly with 0.2 pmol of each fragment and NEB HiFi Assembly Master Mix (E2621, NEB) at 50 °C for 60 minutes. The library is then bottlenecked by taking 300 amol of material from the Gibson assembly reaction (assuming 100% efficiency) and PCR amplifying for 25 cycles with Q5 polymerase and the outnest P5 and P7 primers. The PCR product was run on a 1% agarose gel and a 1.2 kb band was gel extracted, purified, and quantified initially by NanoDrop and subsequently by qPCR (NEBNext Library Quant Kit, E7630,;NEB).

The quantified library was diluted to 2 nM before being denatured (10 µl of library is mixed with 10 µl of 100 mM NaOH and incubated at RT for 5 minutes) and rapidly diluted to 20 pM in HT1 buffer provided by the rapid PE flow-cell clustering kit (PE-402-4002; Illumina). We diluted the library to a final concentration of 6 pM before loading on the HiSeq 2500 and setting up a deep-screening experiment as described above.

Following acquisition of the baseline flow-cell images, we performed an equilibrium-binding assay at successive and increasing concentrations of human HER2–biotin (HE2-H822R, 25 µg; Acro Biosystems) pre-complexed with AF532–SA (S11224; Thermo Fisher) in a 1:1 ratio (100 pM, 333 pM, 1 nM, 3.33 nM, 10 nM, 33.3 nM and 100 nM). In this instance, a binding assay cycle was conducted by injecting 120 µl of the HER2–biotin and AF532–SA pre-complex, incubating for 45 minutes at 20 °C and washing with 200 µl of display buffer before imaging the flow cell. Following the highest 100 nM condition, a kinetic dissociation assay was conducted by pumping display buffer over the flow cell and imaging at 5 minutes, 10, minutes 20, minutes 60 minutes, 120 minutes and 240 minutes. Images were then processed, and CDR sequences were resolved through internal primer sequencing as described above, which we used to assemble a CDR-binding dataset termed HER2affmat.

## BERT-DS architecture and training

ML models have been widely applied to protein engineering[6]. For antibody engineering, we built BERT-DS with inspiration from Prot-Bert[7,8], but with substantially fewer parameters and more focused towards learning the general rules that govern antibodies. Using the PyTorch framework (v.1.8.0) (refs. 9,10), we initially constructed a BERT masked language model (MLM) with a positional embedding input with a vocabulary size of 25 and a maximum input length of 150, as previously described[11]. The positional embedding layer output is then passed into 12 self-attention transformer blocks, with each layer containing 12 attention heads, with a feed-forward layer size of 768, a position-wise feed-forward layer size of 3,072 and dropout of 0.1. The last transformer block output is then passed to the MLM block, which consists of a fully connected layer with a dimension of 768 followed by a Tanh activation function, another fully connected layer with a dimension of 768 followed by a layer normalization[12], and a Gaussian Error Linear Unit (GELU) activation function and a final fully connected layer that produces a 150 × 25 logits matrix. This yielded a model comprising 86 million parameters (Extended Data Fig. 4 for a visual structure of the model).

A total of 20 million human sequences from the Observed Antibody Space (OAS) dataset[13,14] were prepared by downloading the unpaired dataset on 1 August 2021 and extracting all human VH sequences shorter than 150 AAs stored in CSV files. This yielded 229 million unique sequences that we shuffled and split. The first 20 million sequences were used as a training set and the last 100,000 sequences were used as a validation set. For input into the BERT-DS model, each sequence or sample is processed such that, at every AA position, there is a 15% probability that it is either masked (with an 80% probability) or randomly mutated (with a 10% probably), with a subsequent 10% probability that the position is unchanged. The modified and ground-truth sequences are then tokenized for input to the BERT-DS model and padded out to a maximum length of 150 using a padding token.

BERT-DS was pre-trained against our OAS dataset with a cross-entropy loss function and the Adam optimizer using a learning rate of $1 \times 10^{-4}$ and default hyperparameters ($\beta = (0.9, 0.999)$, $\varepsilon = 1 \times 10^{-8}$ and weight decay = 0), 16,000 warm-up steps over which the learning rate is ramped from 0 to $1 \times 10^{-4}$, before being ramped back to 0 over 100,000 optimizer steps; where $\beta$ are coefficients used for computing running averages of the gradient and its square, and $\varepsilon$ is a term added to the denominator of the optimiser function to improve numerical stability. This was conducted through model parallelism on 8 Nvidia A100 GPUs.

Following pre-training, we next wanted to fine-tune the model to perform classification of anti-HER2 VH sequences using data generated by deep screening as a part of the HER2affmat dataset. This was achieved by using the pre-trained embedding and transformer blocks and taking the mean across the vocabulary vector from the last transformer block, yielding a 768-long vector. We then appended a classifier head, which consists of a fully connected layer with an output dimension of 128 followed by a Rectified Linear Unit (ReLU) activation function and a final fully connected layer that outputs logits for 3 classes (non-hit, low hit and high hit). We trained this using a cross-entropy loss with class weights to help training on a heavily imbalanced problem (232,693 non-hit, 1,284 low hit and 111 high hit), and the Adam optimizer with a learning rate of $1 \times 10^{-4}$ and default hyperparameters ($\beta = (0.9, 0.999)$, $\varepsilon = 1 \times 10^{-8}$ and weight decay = 0). No scheduler or warm-up steps were used. We trained the model for 100 epochs and used the checkpoint with the lowest-scoring validation set cross-entropy loss for subsequent predictions.

Fine-tune training used the HER2affmat dataset to model a sequence–function relationship. As we only sequenced the VH3 of each clone, we assumed that the rest of the scFv was wild type. We compose a full VH sequence, tokenize and pad this for input into the BERT-DS

model. For the class labels, we decided on three classes, non-hit, low hit and high hit, that we wished to predict. Each sequence was given a label based on its $FI_{mean}$ in the 5 minute wash condition. We selected threshold values such that the parent clone G98A was roughly centred in the low-hit category, where G98A has an intensity of 190.05, and we chose a non-hit threshold of 150.0 and a high-hit threshold of 250. The dataset was shuffled and split into train and test sets with a 90:10 ratio before training as described above.

### ML versus random library preparation and deep screening

We devised a selection scheme (Fig. 5b) where for each seed sequence a random mutation set was compiled from all single mutants and up to 1,000 mutants from edit distances 2–5, yielding a pool of 13,121 mutations (random/mut). We next assembled a pool of sequences with exclusively ML-generated mutations by removing all sequences with a high-hit score <0.9 and randomly selecting up to 1,000 mutants from edit distances 2–5 and rejecting those that were already selected in the random/mut set. This assembled a pool of 11,916 mutations (ml/mut). Sequences were combined into an oligonucleotide pool of 25,042 CDR VH3 sequences and ordered from Twist Bioscience. The HER2 ML versus random library was assembled for deep screening similar to the HER2affmat library, where 20 cycles of PCR using Q5 polymerase, the upstream and downstream fragments of G98A, were combined with the oligonucleotide pool and the G98A olap forward and reverse primers (Supplementary Table 1). This product was then column purified using a PCR clean-up kit (T1030S; NEB). We next append the deep-screening 5′ and 3′ adaptors using the Gibson assembly with 0.2 pmol of each fragment and NEB HiFi Assembly Master Mix (E2621; NEB) at 50 °C for 60 minutes. The library is then bottlenecked by taking 300 amol of material from the Gibson assembly reaction (assuming 100% assembly efficiency) and PCR amplifying for 25 cycles with Q5 polymerase and the outnest P5 and P7 primers. The PCR product was run on a 1% agarose gel and a 1.2 kb band was gel extracted, purified, and quantified initially by NanoDrop and subsequently by qPCR (NEBNext Library Quant Kit, E7630; NEB).

The quantified library was diluted to 2 nM before being denatured (10 µl of library is mixed with 10 µl of 100 mM NaOH and incubated at RT for 5 minutes) and rapidly diluted to 20 pM in HT1 buffer provided by the rapid PE flow-cell clustering kit (PE-402-4002; Illumina). We diluted the library to a final concentration of 6 pM before loading on the HiSeq 2500 and setting up a deep-screening experiment as described above.

Following acquisition of the baseline flow-cell images, we performed an equilibrium-binding assay at successive and increasing concentrations of human HER2–biotin (HE2-H822R, 25 µg; Acro Biosystems) pre-complexed with AF532–SA (S11224; Thermo Fisher) in a 1:1 ratio (100 pM, 333 pM, 1 nM, 3.33 nM, 10 nM, 33.3 nM and 100 nM). In this instance, a binding assay cycle was conducted by injecting 120 µl of the HER2–biotin and AF532–SA pre-complex, incubating for 45 minutes at 20 °C and washing with 200 µl of display buffer before imaging the flow cell. Following the highest 100 nM condition, a kinetic dissociation assay was conducted by pumping display buffer over the flow cell and imaging at 5 minutes, 10 minutes, 20 minutes, 60 minutes, 120 minutes and 240 minutes. Images were then processed, and CDR sequences were resolved through internal primer sequencing as described above, which we used to assemble a CDR-binding dataset termed HER2 ML versus random.

### Anti-HER2 hit kinetics measurements

Kinetics of binding for all anti-HER2 Fabs was measured using Octet BLI and SA-coated tips (18-5136; Sartorius). In all cases, the buffer used was DPBS (14190-169; Gibco) with 0.1% BSA and 0.02% Tween 20. Purified Fabs were diluted to a final concentration of 20 nM. Kinetics were measured using the following steps: (1) sensor check for 60 seconds, (2) loading of human HER2–biotin (HE2-H822R, 25 µg; Acro Biosystems) at 5 µg ml⁻¹ for 30 seconds, (3) baseline measurement for 60 seconds,

(4) association kinetics at 20 nM of each Fab for 300 seconds, and (5) dissociation kinetics for 600 seconds in buffer.

### Ablation study of BERT-DS

With experimentally validated performance comparisons of BERT-DS against random mutagenesis, we sought to conduct an ablation study to understand whether BERT-DS provides an improvement over BERT-DS without pre-training and over a variety of classical ML models (including a multi-layered perceptron (MLP), linear regression, linear support vector machine (SVM) and random forest models).

As BERT-DS was initially trained in August 2021 and this ablation study was performed in March 2023, considerable changes to our software and hardware stack have affected our ability to fairly compare BERT-DS as described above, with the only exception being the absence of early stopping. Therefore, we decided to set a static train–test split and random seed before (re)training all models with the exact same hardware and software.

The train–test split contained the following counts: for training, 149,402 (non-hit), 899 (low hit) and 81 (high hit); for testing, 37,328 (non-hit), 245 (low hit) and 22 (high hit).

**A randomly initialized BERT-DS.** To understand whether pre-training a language model on an antibody-specific dataset provides benefit to downstream antibody-affinity predictions, we randomly initialized a BERT-DS model and proceeded with the fine-tuning process as described above.

**A soft classification target for BERT-DS.** During development of BERT-DS, we wondered whether converting a continuous set of values with some degree of experimental noise into a multi-class classification problem (non-hit, low hit and high hit) was the cause of our poor $F_1$ scores on the high-hit class. We hypothesized that hard boundaries between the classes and small sample numbers were resulting in a high hit being classified as a low hit and vice versa when such a hit had an FI value close to the boundary. Therefore, we implemented the idea of a soft classification target, where we treat the task of predicting whether a given sequence is a non-hit, low hit or high hit as a binary classification problem; that is, non-hits have a label of 0.0, low hits have a label of 0.5 and high hits have a label of 1.0. We implemented this training regime for a pre-trained and randomly initialized BERT-DS model.

**MLP.** To explore the utility of a large language model over an MLP neural network, we implemented a simple multiple-layered, fully connected neural network. The input to this model is the 21-AA-long CDR3 loop. This is converted into a 64-dimensional feature vector using an embedding layer, which is flattened along the sequence dimension. We then have one linear layer that receives a 21 × 64 dimensional input to a 32-long vector without a bias. This is followed by a GELU activation function and layer normalization. This is followed by another linear layer that increases from 32 dimensions to 64 without a bias, followed again by a GELU and a layer norm. Finally, the last linear layer takes a 64-dimensional input and outputs either a 3D output for multi-class classification or a single scalar value for predicting a soft classification target (as described above). This leads to 46,979 and 46,849 parameters, respectively. Training of both MLP models was performed until convergence with a batch size of 200 on 4× A100 GPUs. Losses of non-hit instances were downweighed by a factor of 1–99 and losses were optimized using AdamW with a linear decay from $5 \times 10^{-5}$ to 0 over 50,000 steps.

**Classical ML models.** We also compared BERT-DS against several classical ML models for multi-class classification, provided by the Scikit-learn package (v.1.2.1). This included a linear regression, linear SVM and random forest model. These models were trained using default hyperparameters on the same train–test split as BERT-DS and

a 1D (flattened) one-hot encoding of the AA sequences, as described in the PCA methods.

**Results and discussion.** In this ablation study, we sought to understand the impact of BERT-DS pre-training and its soft classification target on downstream antibody-affinity predictions. We also compared the performance of BERT-DS against a randomly initialized BERT-DS, MLP, and several classical ML models, including logistic regression, linear SVM and random forest.

As considerable time had passed since we first trained and evaluated BERT-DS, we needed to ensure that comparisons were conducted fairly, using the same train–test data, random seeds, hardware and software stack. We first retrained BERT-DS and on the test set observe $F_1$ score increases in the low-hit class (0.33 to 0.39) and $F_1$ score decreases (0.48 to 0.42, $\Delta = -0.06$) when compared with our 2021 BERT-DS model. We suspect that the variations between the two BERT-DS models (2021 and 2023) are due to a combination of the test–train split in 2023 containing fewer high-hit class samples and the fact that we did not use early stopping in 2023, but rather ran the classification training for 100 epochs. Regardless, we will consider BERT-DS 2023 as the benchmark going forward (Supplementary Table 4).

During the development of BERT-DS, we struggled to improve the $F_1$ scores of the low hit and high hit and wondered whether this was the result of binning FI values into three classes; that is, variances on samples at the class boundaries results in misclassifications from the model. Therefore, we decided to explore that concept of a soft classification target, where we treat the task as a binary classification problem, so that non-hits have a value of 0.0, low hits have a value of 0.5 and high hits have a value of 1.0. Although this improved the low-hit $F_1$ score by 4.8%, the high-hit $F_1$ score dropped by 26.5% (Supplementary Tables 5 and 13). We are not entirely sure why this approach negatively affected the BERT-DS model so much, but future work should explore alternative approaches to 'softening' the binned classification tasks.

We next explored whether pre-training a language model on the observed antibody space provided an advantage over randomly initialized weights. After fine-tuning a randomly initialized BERT-DS, we observed a 7.61% increase in the low-hit $F_1$ score and an 8.11% decrease in the high-hit $F_1$ score (Supplementary Tables 6 and 13). Implementing a soft classification target on a randomly initialized BERT-DS resulted in a 13.71% decrease in the low-hit $F_1$ score and a 33.65% decrease in the high-hit $F_1$ score (Supplementary Tables 7 and 13). Taken together, we show that pre-training on a domain-specific dataset provides a substantial improvement in performance over random initialization (Supplementary Table 13), an effect that was observed with natural-language tasks in the original BERT article[15].

Following our explorations into ablating BERT-DS, we wanted to evaluate prediction performance using an MLP, logistic regression, SVM and random forest. Pre-training and fine-tuning BERT-DS is computationally intensive relative to these simpler models, and we wanted to understand whether this is a justified cost. The results here indicate that BERT-DS, particularly the pre-trained and soft classification target variant, outperforms all classical ML models in terms of $F_1$ scores for all classes in the test set (Supplementary Tables 8–13 and Extended Data Fig. 5). This finding shows that the use of pre-trained language models, specifically tailored to the domain of interest, can provide substantial benefits for antibody-affinity prediction.

**Reporting summary**

Further information on research design is available in the Nature Portfolio Reporting Summary linked to this article.

## Data availability

The main data supporting the results in this study are available in summarized form from Zenodo at https://doi.org/10.5281/zenodo.8241732, under a CC-BY-NC-ND licence. The raw datasets generated during the study are too large to be publicly shared, yet these and other raw data are available from the corresponding author on reasonable request. Source data are provided with this paper.

## Code availability

The custom code for the processing of images collected in a deep-screening experiment and training of BERT-DS is available at https://github.com/holliger-lab/DeepScreening under a CC-BY-NC-ND licence.

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

## Acknowledgements

We thank our colleagues J. Attwater, G. Houlihan, E. Derivery and T. Stevens. We thank our collaborators C. Tuckey, P. Coupland and J. Hadfield for their helpful advice and support throughout the course of this work. We thank the Biological Expression Team (BET) at AstraZeneca, specifically N. Hudson and S. Javed for their assistance in Fab expression and purification. We also thank the following LMB facilities and staff for their general support of this work: LMB Scientific Computing (J. Grimmett, T. Darling and I. Clayson), LMB Mechanical Workshop (S. Scotcher and A. Fowle), LMB Electronics Workshop (A. Howe), LMB Instrument Services (P. Marriott, D. Few and J. Ingle), LMB Biophysics Facility (S. McLaughlin and C. Johnson), LMB Light Microscopy Facility (N. Barry, J. Howe, J. Boulanger, B. Sutcliffe and J. Manton) and LMB Mass Spectrometry Facility (H. Kramer, M. Skehel and S. Maslen). This work was supported by the Medical Research Council (MRC) grant program number MC_U105178804 (P.H.), a research collaboration between AstraZeneca UK and the MRC— MRC–AstraZeneca Blue Sky Grant (BTP), and by an EMBO Long-term Postdoctoral Fellowship ALTF 648-202 (to M.J.L.J.F.).

## Author contributions

B.T.P. and P.H. conceived the concept and designed all the experiments. B.T.P. performed all experiments, except nanobody selections (performed by M.B.) and expression (by M.B. and M.J.L.J.F.); anti-IL-7 antibody library, expression, purification and characterization (by R.M., G.B. and A.R.); and anti-HER2 antibody expression, purification and characterization (by G.B. and A.R.). A.B., R.M. and T.V. enabled and supported the studies on anti-IL-7 and anti-HER2 antibody characterization and validation. B.T.P. wrote all data-processing and analysis software, with support from M.V. B.T.P. and K.J. developed, trained and evaluated the ML model, and performed the ablation study. All authors analysed the data, discussed the results and co-wrote the article.

## Competing interests

The MRC-LMB has filed a patent application on the methodologies described in this article, with B.T.P. and P.H. named as inventors. A.B., G.B. and A.R. are current employees of AstraZeneca. M.B. is a current employee of UCB Pharma. R.M. is a current employee of Alchemab Therapeutics. The other authors declare no competing interests.

## Additional information

**Extended data** is available for this paper at https://doi.org/10.1038/s41551-023-01093-3.

**Correspondence and requests for materials** should be addressed to Philipp Holliger.

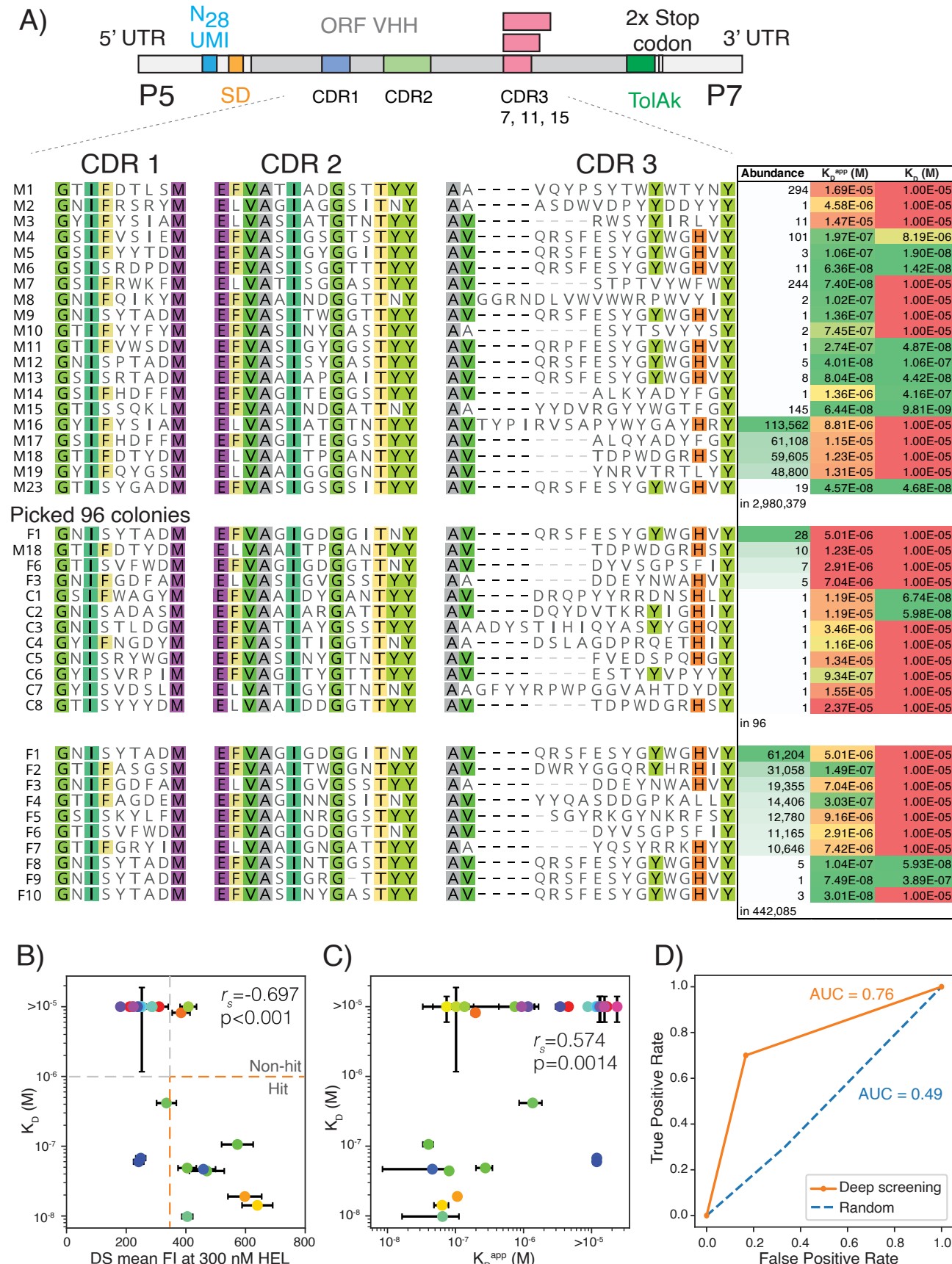

**Extended Data Fig. 1 | See next page for caption.**

**Extended Data Fig. 1 | Characterization of anti-HEL nanobodies. a)** Anti-HEL nanobody hit candidates selected for characterisation, showing the library construct structure (top) and clone ID, where M1-23 are derived from the R3 MACS library, C1-8 were identified by colony picking from the R3 MACS output and F1-10 were derived from the R3 FACS library. The abundance, CDR sequences, a deep screening derived equilibrium binding constant ($K_D^{app}$), and BLI derived kinetic $K_D$s are also shown. **b)** BLI $K_D$s plotted against deep screening FI at 300 nM HEL for all characterised clones, revealing a Spearman's rank correlation constant ($r_s$) of -0.697 and a p-value (determined by two-tailed test) <0.001. Error bars are the errors from fitting respective binding constants. Hit thresholds are shown as dashed orange and grey lines. **c)** BLI $K_D$s plotted against deep screening $K_D^{app}$s for all characterised clones revealing a Spearman's rank correlation constant ($r_s$) of 0.574 and p-value of 0.0014. Error bars are the errors from fitting respective binding constants. **d)** Receiver Operating Characteristic (ROC) curve showing the performance of deep screening at picking hits versus non-hits in a binary classification scheme, and how this compares to random. This curve uses the following hit thresholds: a mean FI at 300 nM HEL of 347.58 and a BLI $K_D$ of $10^{-6}$M. Area under the curve (AUC) values are indicative of performance, with deep screening having an AUC of 0.76, while random is 0.49.

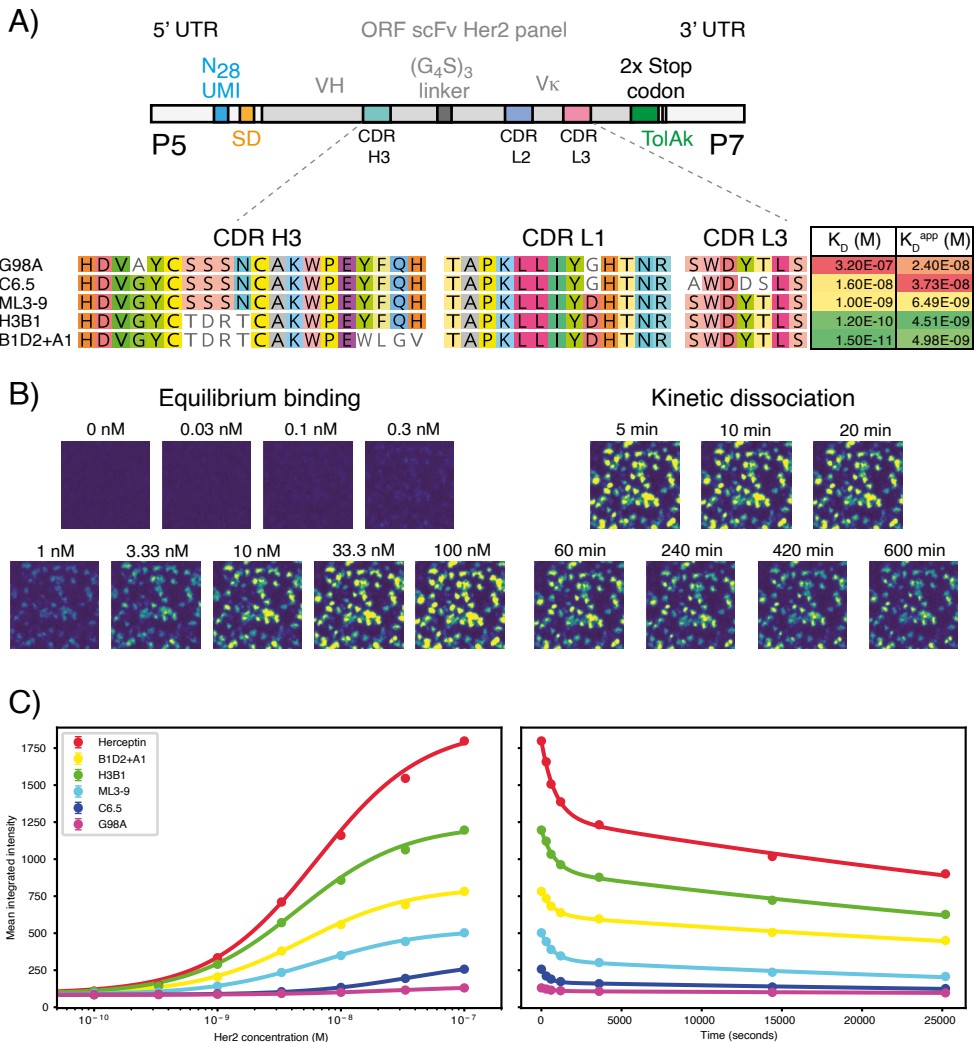

**Extended Data Fig. 2 | Display of an anti-HER2 scFv affinity panel. a)** Construct design showing CDR sequences (VH3, VL1 and VL3) and binding affinities of clones G98A, C6.5, ML3-9, H3B1 and B1D2 + A1. **b)** Flow cell images of the ribosome displayed anti-HER2 scFv affinity panel during equilibrium binding and kinetic dissociation. Images are set to the same min/max threshold of 100/1000. **c)** Curve fits to equilibrium binding and kinetic dissociation data, showing clones from A) and the addition of Herceptin (trastuzumab).

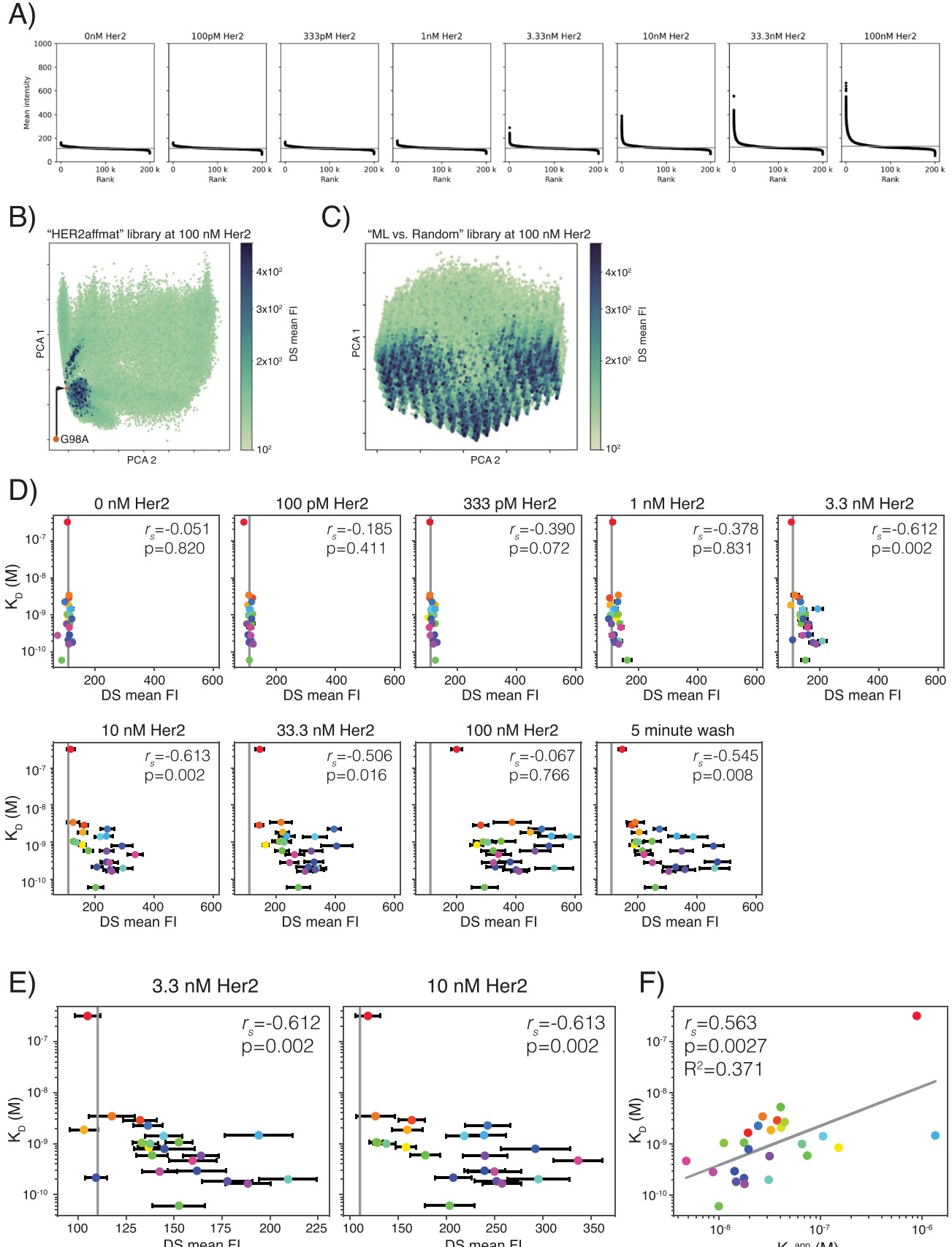

**Extended Data Fig. 3 | See next page for caption.**

**Extended Data Fig. 3 | Rank and correlation plots from deep screening for the HER2 ML vs. Random library. a)** Rank plots of 199k anti-HER2 scFv clones from the equilibrium binding assay, showing their mean fluorescent intensities at 0 nM, 0.1 nM, 0.3 nM, 1 nM, 3.3 nM, 10 nM, 33.3 nM and 100 nM Her2. Clones were selected for a variety of reasons (see Extended Data 1) for subsequent conversion to Fab, expression, purification, and characterisation. **b)** PCA plot from the 'HER2Affmat' library, showing all 236k CDR H3 protein sequences projected into two dimensions and coloured by mean fluorescent intensity at 100 nM of HER2. A red dot shows the position of G98A wild-type relative to the library. **c)** PCA plot from the HER2 ML vs. Random library, showing all 199k CDR H3 protein sequences projected into two dimensions and coloured by mean fluorescent intensity at 100 nM of HER2. **d)** Correlation between BLI characterised binding affinities ($K_D$) and deep screening mean FI at 0 nM, 0.1 nM, 0.3 nM, 1 nM, 3.3 nM, 10 nM, 33.3 nM, 100 nM HER2 and the 5 minute wash condition. Error bars are s.e.m. and $n \geq 12$. The grey vertical line is showing the mean library intensity at each respective concentration. Correlations are shown as Spearman's rank correlation constant ($r_s$) and p-values were determined by a two-tailed test. **e)** Zoomed correlation between BLI characterised binding affinities ($K_D$) and deep screening mean FI at 3.3 nM and 10 nM HER2. Error bars are s.e.m. and $n \geq 12$. **f)** Correlation between BLI characterised binding affinities ($K_D$) and deep screening determined equilibrium binding constants ($K_D^{app}$). Correlations are shown as Spearman's rank correlation constant ($r_s$) and p-values were determined by a two-tailed test. Linear regression was used to show a line of best fit.

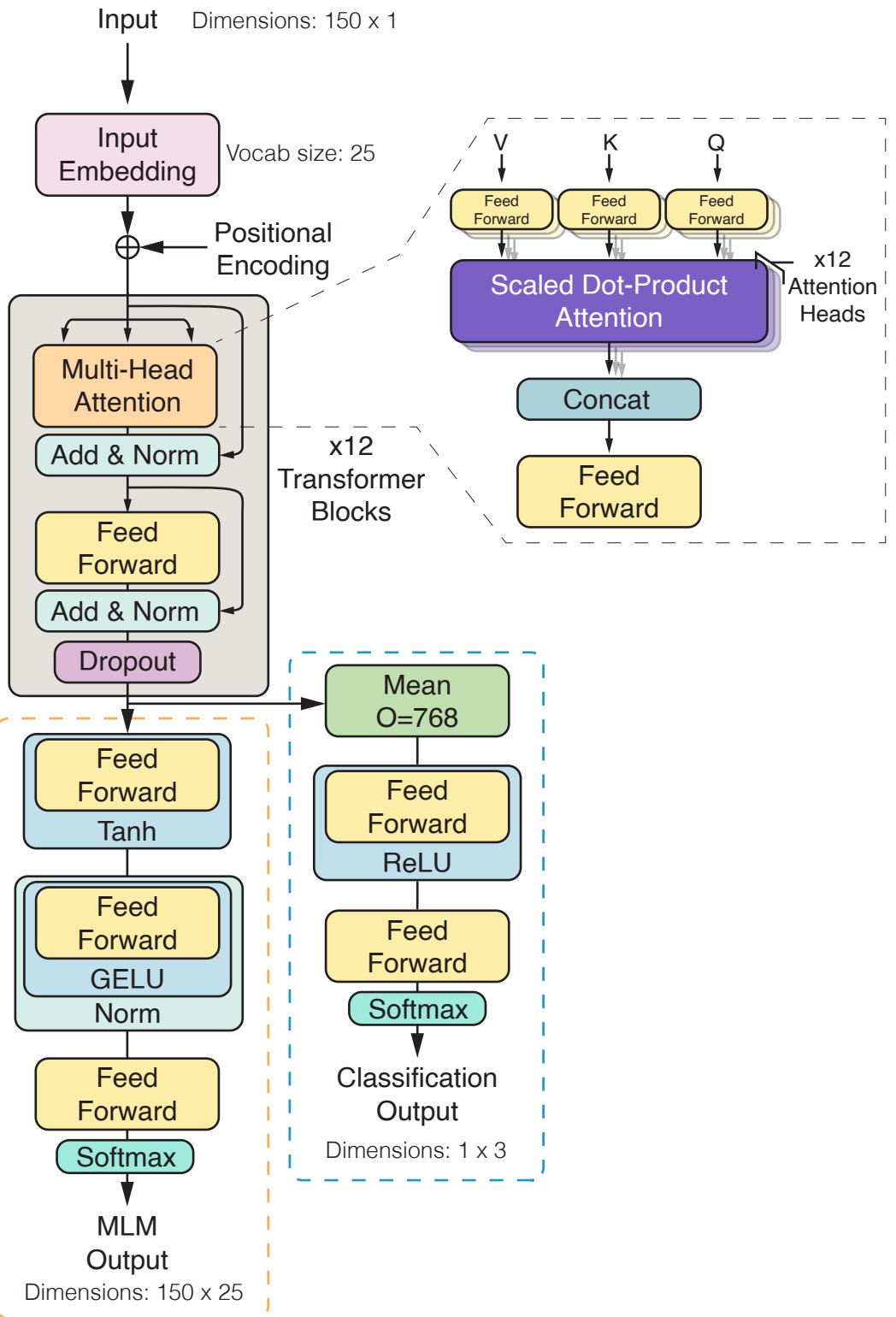

**Extended Data Fig. 4 | Architecture of BERT-DS.** Architecture of the BERT-DS model, showing the input embedding layer, 12 self-attention transformer blocks, the masked language modelling (MLM) output (dashed orange line) and classification output (dashed blue line) heads.

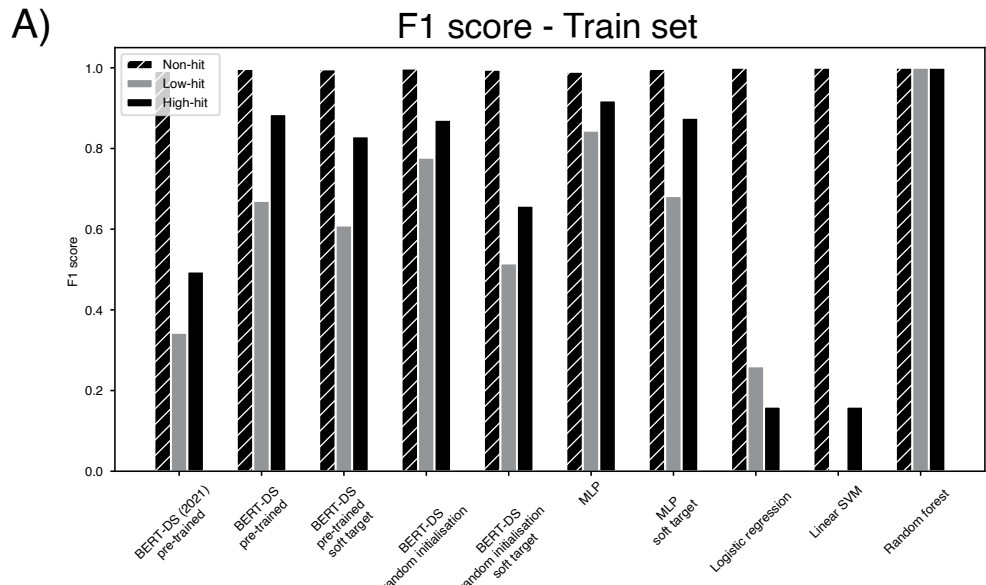

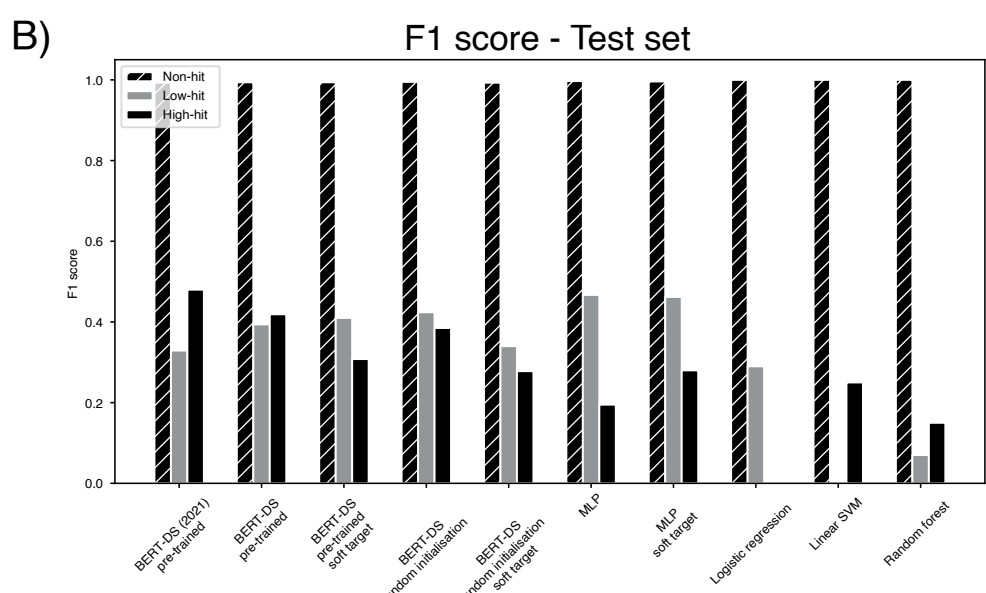

**Extended Data Fig. 5 | Ablation study of BERT-DS.** Ablation study F1 scores on predicting non-hits, low-hits, and high-hits from the 'HER2Affmat' dataset for BERT-DS (with and without pretraining on OAS), BERT-DS ablation models, a multi-layered perceptron (MLP) neural network, an MLP trained on a soft binary classification target, logistic regression, linear support vector machine and random forest models. We report F1 scores on the **a)** train and **b)** test set splits. Numerical values for F1, precision and recall for each model and train/test set can be found in Supplementary Tables 5–14.

# Reporting Summary

## Statistics

For all statistical analyses, confirm that the following items are present in the figure legend, table legend, main text, or Methods section.

| n/a | Confirmed | |
|---|---|---|
| ☐ | ☒ | The exact sample size (*n*) for each experimental group/condition, given as a discrete number and unit of measurement |
| ☐ | ☒ | A statement on whether measurements were taken from distinct samples or whether the same sample was measured repeatedly |
| ☒ | ☐ | The statistical test(s) used AND whether they are one- or two-sided *Only common tests should be described solely by name; describe more complex techniques in the Methods section.* |
| ☒ | ☐ | A description of all covariates tested |
| ☒ | ☐ | A description of any assumptions or corrections, such as tests of normality and adjustment for multiple comparisons |
| ☐ | ☒ | A full description of the statistical parameters including central tendency (e.g. means) or other basic estimates (e.g. regression coefficient) AND variation (e.g. standard deviation) or associated estimates of uncertainty (e.g. confidence intervals) |
| ☒ | ☐ | For null hypothesis testing, the test statistic (e.g. *F*, *t*, *r*) with confidence intervals, effect sizes, degrees of freedom and *P* value noted *Give P values as exact values whenever suitable.* |
| ☒ | ☐ | For Bayesian analysis, information on the choice of priors and Markov chain Monte Carlo settings |
| ☒ | ☐ | For hierarchical and complex designs, identification of the appropriate level for tests and full reporting of outcomes |
| ☐ | ☒ | Estimates of effect sizes (e.g. Cohen's *d*, Pearson's *r*), indicating how they were calculated |

*Our web collection on statistics for biologists contains articles on many of the points above.*

## Software and code

Policy information about availability of computer code

| | |
|---|---|
| Data collection | HiSeq Control software (HCS v. 2.2.68, Illumina). Archimedes Test software (ATS v. 3.8.317.0, Illumina). Octet Data Acquisition software (v. 11.0, ForteBio). |
| Data analysis | Data were analysed using custom software, available at https://github.com/holliger-lab/DeepScreening |

For manuscripts utilizing custom algorithms or software that are central to the research but not yet described in published literature, software must be made available to editors and reviewers. We strongly encourage code deposition in a community repository (e.g. GitHub). See the Nature Portfolio guidelines for submitting code & software for further information.

## Data

Policy information about availability of data

All manuscripts must include a data availability statement. This statement should provide the following information, where applicable:

- Accession codes, unique identifiers, or web links for publicly available datasets
- A description of any restrictions on data availability
- For clinical datasets or third party data, please ensure that the statement adheres to our policy

The main data supporting the results in this study are available in summarized form from Zenodo at https://doi.org/10.5281/zenodo.8241732, under a CC-BY-NC-

## Human research participants

Policy information about studies involving human research participants and Sex and Gender in Research.

| | |
|---|---|
| Reporting on sex and gender | The study did not involve human participants. |
| Population characteristics | — |
| Recruitment | — |
| Ethics oversight | — |

Note that full information on the approval of the study protocol must also be provided in the manuscript.

# Field-specific reporting

Please select the one below that is the best fit for your research. If you are not sure, read the appropriate sections before making your selection.

☒ Life sciences          ☐ Behavioural & social sciences          ☐ Ecological, evolutionary & environmental sciences

For a reference copy of the document with all sections, see nature.com/documents/nr-reporting-summary-flat.pdf

# Life sciences study design

All studies must disclose on these points even when the disclosure is negative.

| | |
|---|---|
| Sample size | Sample sizes for the deep-screening experiments were initially determined by Illumina sequencing and refined via UMI replicate filtering, median absolute deviation outlier rejection, and filters on whether data were observed during the binding assay. |
| Data exclusions | In processing the deep-screening data, measurements were grouped by UMI, and outliers removed by median absolute deviation of 2 standard deviations from the sample median. |
| Replication | The deep-screening experiments were not replicated, although replication studies have been performed outside of this work. Within a given deep-screening experiment, we require that each UMI has at least 12 replicate measurements at different locations on the flow-cell surface. Otherwise a UMI is excluded from further analysis.<br><br>The BLI experiments were not replicated.<br><br>TF-1 STAT-5 IL7 receptor-signalling assays were performed in duplicate for each anti-IL7 Fab that was produced. |
| Randomization | No randomization outside of random mutagenesis (experimental/in silico) was performed or required. |
| Blinding | Blinding was not applicable to this study, because data acquisition and quantification were performed by machines. |

# Reporting for specific materials, systems and methods

We require information from authors about some types of materials, experimental systems and methods used in many studies. Here, indicate whether each material, system or method listed is relevant to your study. If you are not sure if a list item applies to your research, read the appropriate section before selecting a response.

### Materials & experimental systems

| n/a | Involved in the study |
|---|---|
| ☐ | ☒ Antibodies |
| ☒ | ☐ Eukaryotic cell lines |
| ☒ | ☐ Palaeontology and archaeology |
| ☒ | ☐ Animals and other organisms |
| ☒ | ☐ Clinical data |
| ☒ | ☐ Dual use research of concern |

### Methods

| n/a | Involved in the study |
|---|---|
| ☒ | ☐ ChIP-seq |
| ☐ | ☒ Flow cytometry |
| ☒ | ☐ MRI-based neuroimaging |

# Antibodies

| | |
|---|---|
| Antibodies used | FITC labelled anti-HA antibody (GG8-1F3.3.1, Miltenyi Biotech). |
| Validation | The manufacturer states that 293HEK cells transiently transfected with HA-tagged CD4 were stained intracellularly with Anti-HA-FITC and CD4-PE, and analysed by flow cytometry to control gene-of-interest and MACSelect surface-marker expression. <br> The manufacturer shows a flow-cytometry plot that indicates the validation of the product. <br> https://www.miltenyibiotec.com/GB-en/products/ha-antibody-gg8-1f3-3-1.html#fitc:30-tests-in-60-ul |

# Flow Cytometry

## Plots

Confirm that:

☒ The axis labels state the marker and fluorochrome used (e.g. CD4-FITC).

☒ The axis scales are clearly visible. Include numbers along axes only for bottom left plot of group (a 'group' is an analysis of identical markers).

☒ All plots are contour plots with outliers or pseudocolor plots.

☒ A numerical value for number of cells or percentage (with statistics) is provided.

## Methodology

| | |
|---|---|
| Sample preparation | The nanobody yeast-display library was acquired from the Kruse laboratory as a frozen stock of >2.5x10^9 cells (EF0014-FP, Kerafast). To prepare the naïve library for the first round of selection, one aliquot was thawed at 30°C and used to inoculate 1 L of Yglc4.5 –Trp supplemented with 2% galactose. The culture was then grown for 72 hours at 24°C. Expression was confirmed by flow cytometry with a FITC-labelled anti-HA antibody (GG8-1F3.3.1, Miltenyi Biotech) prior to the first round of selection. Round 3 FACS was conducted by incubating cells with 200 nM HEL-biotin for one hour at 4°C, pelleted and resuspended in fresh PBS-T-BSA and combined with 100 µg of Neutravidin-PE (A2660, ThermoFisher Scientific) and a 1:1000 dilution of the anti-HA-FITC (GG8-1F3.3.1, Miltenyi Biotech) antibody for 15 minutes before being sorted. |
| Instrument | Synergy 3 cell sorter (Sony Biotechnology). |
| Software | SY3200 software v2.0 (Sony Biotechnology). |
| Cell population abundance | 50,135 cells with dual labelled (FITC/PE). |
| Gating strategy | Cells were sorted by gating for dual labelled (FITC/PE) events, yielding 50,135 cells. |

☒ Tick this box to confirm that a figure exemplifying the gating strategy is provided in the Supplementary Information.

