## [Peer Review File · Nature Biomedical Engineering]

Rapid discovery of high-affinity antibodies via massively parallel sequencing, ribosome display, and affinity screening

Corresponding author: Philipp Holliger

Editorial note

This document includes relevant written communications between the manuscript's corresponding author and the editor and reviewers of the manuscript during peer review. It includes decision letters relaying any editorial points and peer-review reports, and the authors' replies to these (under 'Rebuttal' headings). The editorial decisions are signed by the manuscript's handling editor, yet the editorial team and ultimately the journal's Chief Editor share responsibility for all decisions.

Any relevant documents attached to the decision letters are referred to as **Appendix #**, and can be found appended to this document. Any information deemed confidential has been redacted or removed. Earlier versions of the manuscript are not published, yet the originally submitted version may be available as a preprint. Because of editorial edits and changes during peer review, the published title of the paper and the title mentioned in below correspondence may differ.

Correspondence

Wed 25 Jan 2023

Decision on Article nBME-22-3040-T

Dear Prof Holliger,

Thank you again for submitting to *Nature Biomedical Engineering* your manuscript, "Rapid discovery of high-affinity antibodies by deep screening". The manuscript has been seen by three experts, whose reports you will find at the end of this message.

You will see that two of the reviewers appreciate the work. However, they express concerns about the degree of support for the claims, and provide useful suggestions for improvement. Editorially, we have placed emphasis on the throughput and speed advantages of the method (rather than on its perceived conceptual novelty), and hence we require strong support for the stated practical and performance advantages. We hope that with significant further work you can address the criticisms and convince the reviewers of the merits of the study. In particular, we would expect that a revised version of the manuscript provides:

- * Thorough statistical analyses of the reported correlations for antibody–antigen binding.
- * Extended benchmarking of the performance of the BERT model.
- * Clearer and balanced discussion of the advantages and current limitations of the technology.

When you are ready to resubmit your manuscript, please upload the revised files, a point-by-point rebuttal to the comments from all reviewers, the reporting summary, and a cover letter that explains the main improvements included in the revision and responds to any points highlighted in this decision.

Please follow the following recommendations:- * Clearly highlight any amendments to the text and figures to help the reviewers and editors find and understand the changes (yet keep in mind that excessive marking can hinder readability).
- * If you and your co-authors disagree with a criticism, provide the arguments to the reviewer (optionally, indicate the relevant points in the cover letter).
- * If a criticism or suggestion is not addressed, please indicate so in the rebuttal to the reviewer comments and explain the reason(s).
- * Consider including responses to any criticisms raised by more than one reviewer at the beginning of the rebuttal, in a section addressed to all reviewers.
- * The rebuttal should include the reviewer comments in point-by-point format (please note that we provide all reviewers will the reports as they appear at the end of this message).
- * Provide the rebuttal to the reviewer comments and the cover letter as separate files.

We hope that you will be able to resubmit the manuscript within 15 weeks from the receipt of this message. If this is the case, you will be protected against potential scooping. Otherwise, we will be happy to consider a revised manuscript as long as the significance of the work is not compromised by work published elsewhere or accepted for publication at *Nature Biomedical Engineering*.

We hope that you will find the referee reports helpful when revising the work. Please do not hesitate to contact me should you have any questions.

Best wishes,

Pep

Pep Pàmies
Chief Editor, Nature Biomedical Engineering

Reviewer #1 (Report for the authors (Required)):

Here the authors leverage next-generation sequencing (NGS) technology for the high-throughput sequencing, protein display, and functional screening of antibody libraries. Distinct from classic protein display technologies such as yeast and phage display where variants with a specific desired phenotype are selected and then sequenced, here NGS is directly linked with functional screening in a more expeditious process. As the authors acknowledge, a similar approach to screen for protein-protein interactions has been described in previous work; however, the approach described herein (“deep screening”) allowed the functional screening of antibody libraries on the order of 10⁸ variants. Finally, the authors use their deep screening data to train machine learning models able to generate novel antibody sequence candidates likely to have high-affinity for their target.

1. As the authors note, previous work has performed protein synthesis and display on the Illumina platform (PMC5183537, PMC7001660, ...). The authors should articulate more clearly how deep screening is fundamentally distinct from this previous work, other than by increasing the throughput.

2. In the Introduction, the authors state that classic display technologies such as yeast and phage display suffer from limitations such as inherent biases and an ability to only sample a fraction of the possible sequence space – implying that deep screening is an improvement over these technologies as the method does not have these same limitations. However, it would seem that biases due to differences in expression and protein folding would still impact signal intensity on the flow cell (apparent affinity). Furthermore, although deep screening is able to interrogate on the order of 10⁸ variants, this is still only a fraction of the

possible sequence space. Therefore, authors should revise the manuscript text to describe deep screening in the context of the overall theoretical protein sequence space.

3. In the proof-of-concept screen on the yeast-surface display nanobody (VHH) library, 2-fold over library background is considered a positive signal. Why is above this threshold considered bona fide HEL binding? Also, it is not clear what the library background is; it would be helpful to state more explicitly what this means.

4. The authors present several correlations without the significance clearly stated, making data interpretation challenging. It would be helpful if the authors added p-values to these plots and mentioned in the text whether the correlations were statistically significant or just a trend.

5. The authors suggest at several points in the manuscript that the results generated via deep screening are precise enough to accurately rank clones by their binding ability. However, it is unclear how effectively deep screening is able to do this. For example, while there is a relatively strong correlation between deep screening binding and BLI-measured binding in figure S2B, this trend appears to be largely driven by the negatives. If the negatives were removed from this plot, there would likely be minimal correlation between the deep screening- and BLI-determined affinities. Consistent with this notion, in figure S11A, deep screening data provided low resolution with respect to differences in binding between the group of high-affinity clones (ML3-9, H3B1, and B1D2+A1) and between the group of low-affinity clones (G98A and C6.5). This limited ability of deep screening to accurately rank clones by their binding ability may underlie the relatively poor performance (F1-scores) of their BERT-DS model in identifying “low-hit” and “high-hit” sequences. Thus, while the authors seem correct in their assertion that deep screening can effectively separate binders from non-binders and identify high-affinity clones, it is not clear whether deep screening offers a high enough level of resolution to faithfully rank clones by binding affinity. This will require the authors to revise their statements / conclusions to be more accurate based on the actual data.

6. The explanation of the BERT-DS machine learning model is inaccessible for a non-expert. The authors should explain what the model is and why it was selected for this particular task to frame the work. Why did the authors not consider other machine learning models / approaches?

7. Evaluating the pre-trained BERT-DS model by comparing its ability to predict sequences with better affinity than those in the “HER2affmat” dataset is a relatively low bar given that simple random mutagenesis and screening can complete the same task. Perhaps comparing the performance of the pre-trained BERT-DS model with a model that was not pre-trained, and/or a more simplistic model would more effectively demonstrate the predictive power and value of the pre-trained BERT-DS model.

8. While the performance of BERT-DS is impressive, the authors didn't show any comparisons to more basic, conventional ML models. Is the BERT model necessary to achieve good predictions in a classifier?

9. Can the authors explain why they trained BERT-DS on the full VH dataset available on OAS, rather than human only sequences? As presumably those would allow the model to design much more relevant, therapeutic candidates.

10. The authors show a PCA analysis of the CDR H3 sequence space for the HER2affmat library in Fig 4C. How the PCA analysis was done is not explained in the methods section.

11. Choosing threshold for classifier is an essential step in the BERT-DS model set up. The author mentioned the threshold for high-hit was adjusted in order to have enough high affinity clones, while the ratio of high-hit sequence is still much lower than the other two classes. It will be helpful for others who are interested in implementing this workflow to know how the authors define “enough clone counts” and “balanced class counts”.

Reviewer #2 (Report for the authors (Required)):

Porebski et al present a technically impressive paper in which they repurpose Illumina sequencers to afford high throughput screening of ribosome displayed antibody libraries in a method they call ‘deep screening’. The essence of this technology is the production of RNA islands which then can be in vitro translated with protein captured by ribosome display; using existing Illumina optical detection allows binding measurements

to be performed. The authors demonstrate this technology on identifying rare lysozyme binding clones from partially FACS or MACS enriched VHH library. They then demonstrate this deep screening on two separate affinity maturation campaigns - one against IL-7 (where they could go directly from screening to pM hit identification) and one against Her2. The advantage of this platform is that in a single three day experiment the authors get a pseudo-binary readout of folding/expression/binding (mean FL1; more on this later) as well as sequences for hundreds of thousands of variants. This system affording sequence-function at scale represents a clear advantage for directed evolution affinity maturation campaigns. There are some stones to throw - the initial demonstration of the technology for initial binding discrimination from a library may not represent an advantage compared with further screening rounds by phage/yeast display; the authors, like many contemporaries, use deep learning models when simpler linear models would in fact be superior in reaching the desired answer; the system currently requires single chain or single domain antibodies; the dissociation rate measurements appear poorly correlated with functional antibody properties like dissociation constants. Still, these are manageable concerns and my overall enthusiasm for this paper is high. Given these critiques, the following should be addressed in any revision:

Major issues:

1. For the VHH anti-HEL screening experiment, there is some concern about the correlations reported. In Fig S2, dissociation constants for 25/41 (61%) of clones tested have a BLI-determined K_d of 10 μ M; presumably this is the upper limit of binding - based on the BLI curves presented these appear as non-binders and should be reported as such. I think the appropriate experiment here (can deep screening help identify binders better than randomly picking 96 colonies) should be tested with a binary classification scheme, and statistical significance reported (does deep screening do a better job of hit discrimination than simply plating colonies and screening?). I'm also not positive how to handle the non-binders in the libraries for evaluation of the Spearman Rho - panels 2B, C, but alternative ways of presenting this data should be considered by the authors.

1. (minor related point) - I think the claim 'identification of the same clones by standard procedures would have required either multiple further rounds of selection and/or labour-, cost- and time-intensive microplate screening of thousands of colonies.' is overstated - by FACS you barely have binders when the experiment was stopped, so of course an antibody engineer would have continued with further rounds of yeast screening until robust signals were present. Recommend removing the last phrase from the sentence.

2. For the VHH anti-HEL screening experiment, dissociation constants do a poor job of describing or correlating with the data. From an outside perspective, the Illumina cell is ill-suited for BLI-type measurements as the flow cell can't remove antigen fast enough so you likely have some pseudo-equilibrium dependent on the local environment (antigen can unbind locally and rebind; the extent of local rebinding would be dependent on the presence of strong binder clusters nearby), so the data would be hard to deconvolute. My recommendation is just to remove or move all discussion of this to the supporting information (I understand a ton of work was involved here but it does not help advance the narrative of the technology).

3. Figure 5. The result from affinity maturation by deep learning is overstated. The low hit and high hit categories had high amounts of false positives (F1-scores were low). Furthermore, the comparison group would be confusing to specialists. A very common method for affinity maturation is shuffling or combinatorially sampling mutations from the templated sequence for hits. The three 'seed' sequences reported in Figure 5 (HER2000[3-5]) have 1-5 mutations from the starting variant at 8 positions - the combinatorial library ($2^7 \times 3 = 384$ variants) is quite smaller than either libraries tested and would have reached the exact same answer faster. Inclusion of these language models is en vogue, but the present contribution does not add much to the paper. If the authors insist on keeping this section, then very strong disclaimers must be added in results in discussion to the extent that one hot vector encoding of hits would have reached the same answer much faster and with minimal effort.

Minor issues:

1. Abstract: The practical, demonstrated (not theoretical size limits) should be reported; in the supporting info authors discuss the demonstrated upper limit of 4×10^6 variants (supp page 3), which should be reported.
2. Introduction: The introduction does a reasonable job of surveying the literature; one paper missing is from the Cochran group and should be included and compared with this work: Chen, Bob, et al. "High-throughput analysis and protein engineering using microcapillary arrays." *Nature chemical biology* 12.2 (2016): 76-81.

2. Reporting of error. For Figure 2: → Meaning of error bars and number of replicates should be described in the figure legend. For Figure 3. → 'error bars are 1 s.d., n=2'. It is impossible to record a standard deviation from two data points. Please clarify.

3. For Figure 3, the panel C uses DS FI at 333 pM, but main text states clones were selected by top 19 binding at 1 nM hU1L-7. Please clarify or show the selection data. For the figure it may be appropriate to show a volcano or other plot showing the distribution of DS FI values among clones to show clone selection.

4. Supporting info. The equations listed in the supplemental should be updated to reflect concordance with the main text (in particular K_{Dapp} and k_{offapp}) (page 9; a biexponential fit is performed without elaboration as far as I can ascertain).

Reviewer #3 (Report for the authors (Required)):

In this paper, Porebski et al. provide a platform for fast discovery of binders. While I am sure that speed in discovery is a valuable asset, it is unclear to me how this platform fundamentally differs from other library discovery methods.

The machine learning section reads very antiquated not based on the state-of-the-art. This is probably because the authors go out of their way to almost not cite any paper of relevance from the last years (both from the experimental and computation side of antibody discovery). The authors show a lot of data but these data do not advance our understanding of library methods nor antibody biology.

Thu 01 Jun 2023

Decision on Article nBME-22-3040A

Dear Prof Holliger,

Thank you again for your revised manuscript, "Rapid discovery of high-affinity antibodies by deep screening", which has been seen by the original reviewers. In their reports, which you will find at the end of this message, you will see that the reviewers are essentially happy with the revised manuscript, and that Reviewers #1 and #2 offer a few points that should help you iron out the claims and reporting associated with the machine-learning models.

I am hoping that you will be able to offer satisfactory replies to both reviewers.

As before, when you are ready to resubmit your manuscript, please upload the revised files, a point-by-point rebuttal to the comments from Reviewers #1 and #2, and the reporting summary.

Best wishes,

Pep

Pep Pàmies
Chief Editor, Nature Biomedical Engineering

Reviewer #1 (Report for the authors (Required)):

The authors have sufficiently responded to most of my comments and the revised manuscript has been improved. However there are still two remaining issues related to the following:

1.8 response

The difference in F1 between the pretrained vs. random initialization BERTs trained in 2023 is less than the F1 between pretrained 2023 vs pretrained 2021 models. How much of the difference in performance seen between pre-train and random initialization might be due to train/test set batching? With so much variability, are the authors able to be so certain that pretraining contributes significantly to the performance of the DS-BERT model?

1.11 response

The authors reformatted the classifier structure, however still missing the illustration of why choosing the threshold which yielded 111 high-hit was enough for training and a class count of 232,693 non-hit, 1,284 low-hit and 111 high-hit was considered to be balanced.

Reviewer #2 (Report for the authors (Required)):

The authors have done a fantastic job of responding to most of my concerns in the initial manuscript. Let me reiterate that the platform development and execution is top-rate and there are no technical concerns with the development of the platform, the demonstrations of the technology, or the results presented.

With great respect to the authors, the remaining concern I have is with the overstatement of the DL model presented compared with what a typical protein engineer workflow would comprise for a task-specific engineering exercise (improve the properties of an antibody). Before I go in further detail, let me mention that

I have no technical concerns with the ML portion of the paper and I commend the authors for inclusion of competing ML analysis (MLP, SVN, etc) - this strengthens the paper. So, on balance, I don't need to re-review this paper as it is technically sound and represents an impactful advance. Rather, I'll leave this to the editor's discretion (and, I suppose, the authors). I feel strongly that down-weighting the significance of the ML model is important, and this is where my subjectivity shows - while I am quite bullish on unsupervised learning to eventually learn functional properties of antibodies, my personal opinion is that many overstate the importance of such models for task-specific purposes (improve the properties of antibody 'A'), when simpler models reach the 'best' answer faster and with much less labour. Let me explain my reasoning.

It is well established in directed evolution of biomolecules that if the screen/selection is towards some objective function (higher yields, lower K_m , lower K_d , folding propensity, etc) that the hits resulting from the screen can be combined; this can be through DNA shuffling or through combinatorial library generation through synthetic DNA. The methods are immaterial. What you are doing in practice is sampling within the functional space of the molecules identified from the first round of screening. So, contrary to the claims from the authors response on the combinatorial complexity of sampling, the actual functional space you would seek to sample in practice is much lower. There is a good older literature reference for this in Nature Biotech (ref 1) - the crux of this is a very old supervised learning technique (a form of linear regression!) based on one hot encoding of amino acid positional hits (this is what I was trying to convey in my first review; the authors rightly pushed back as I was imprecise) was fantastic at navigating functional search space.

In this particular example, the authors identified initial hits as seed sequences comprising the following sets of mutations:

98GA
99YQW
101SDT
102ST
103SP
104NRT
106AL
107KT

So, faced with these hits, a protein engineer would encode the combinatorial library encoding diversity at these positions, these positions only, and only these residues. If she/he did, they would end up with the exact same answer (to the question: 'find the best antibody') as the authors did with a fraction of the effort (by only screening through several hundred sequences in this example):

98G
99Y
101S
102T
103S
104R
106L
107T

What if there were more hits than those represented in the seed sequence? In practice, the authors would convert the frequency at each position by one hot encoding and encode combinatorial libraries of the most frequently represented residues at each position in the CDR; since those seed sequences likely represent positions with high frequency in the hits, the answer would likely be contained in the encoded library, again with a fraction of the effort. It would have been fantastic if the language model used by the authors identified residues not present in the seed sequences for the best hit, but this was not the case.

I'm not quite sure how the authors can address this or what suggestions to convey to the editor - the language model used does give additional information (what are sequences not likely to bind the given target) that you can't get with this crude workflow, but for the simpler task of identifying the best antibody sequence other methods are superior in that they would identify the best answer as the language model but with a fraction of the effort. My original review suggested a strong rephrasing of the language stating the concerns noted above; I still believe this for the reasons discussed above. Ultimately, the novelty of the paper is in the assay, not the language model.

References:

1. Fox, R. J., Davis, S. C., Mundorff, E. C., Newman, L. M., Gavrilovic, V., Ma, S. K., ... & Huisman, G. W. (2007). Improving catalytic function by ProSAR-driven enzyme evolution. *Nature biotechnology*, 25(3), 338-344.

Reviewer #3 (Report for the authors (Required)):

The authors have addressed all my concerns.

Tue 06 Jun 2023

Decision on Article nBME-22-3040B

Dear Prof Holliger,

Thank you for the latest version of your manuscript, "Rapid discovery of high-affinity antibodies by deep screening". Having consulted with Reviewers #1 and #2 (whose comments you will find at the end of this message), I am pleased to write that we shall be happy to publish the manuscript in *Nature Biomedical Engineering*.

We will be performing detailed checks on your manuscript, and in due course will send you a checklist detailing our editorial and formatting requirements. You will need to follow these instructions before you upload the final manuscript files.

Best wishes,

Pep

Pep Pàmies
Chief Editor, Nature Biomedical Engineering

Reviewer #1 (Report for the authors (Required)):

The authors have sufficiently answered my questions and I have no additional comments to make at this point.

Reviewer #2 (Report for the authors (Required)):

I thank the authors for the reply and agree with many, though not all, points raised in the rebuttal. However, these differences are ones of opinions, not data, and so I commend the authors for a solid contribution to the literature. Well done.

Rebuttal 1

Porebski et al: Response to the referees' comments and suggestions

Reviewer #1 (Report for the authors (Required)):

Here the authors leverage next-generation sequencing (NGS) technology for the high-throughput sequencing, protein display, and functional screening of antibody libraries. Distinct from classic protein display technologies such as yeast and phage display where variants with a specific desired phenotype are selected and then sequenced, here NGS is directly linked with functional screening in a more expeditious process. As the authors acknowledge, a similar approach to screen for protein-protein interactions has been described in previous work; however, the approach described herein (“deep screening”) allowed the functional screening of antibody libraries on the order of 10⁸ variants. Finally, the authors use their deep screening data to train machine learning models able to generate novel antibody sequence candidates likely to have high-affinity for their target.

1.1. As the authors note, previous work has performed protein synthesis and display on the Illumina platform (PMC5183537, PMC7001660, ...). The authors should articulate more clearly how deep screening is fundamentally distinct from this previous work, other than by increasing the throughput.

In the revised manuscript, we have now expanded the discussion on preceding display approaches on the different Illumina platforms outlining in more detail where deep screening is distinct and offers potential advantages. Specifically, we argue that ca. 100-fold increased throughput / screening depth, covalent RNA linkage to the flow cell and increased DNA / RNA fragment length (enabling scFv display) are both distinctive features and offer important enhancements to the utility of the method.

1.2.1. In the Introduction, the authors state that classic display technologies such as yeast and phage display suffer from limitations such as inherent biases and an ability to only sample a fraction of the possible sequence space – implying that deep screening is an improvement over these technologies as the method does not have these same limitations. However, it would seem that biases due to differences in expression and protein folding would still impact signal intensity on the flow cell (apparent affinity).

Reviewer 1 raises an important point. It is of course correct that no display system can be completely free from biases as for example levels of expression and folding will always impact amounts of functionally displayed protein and thereby bias the repertoire of biomolecules that can actually be functionally interrogated. However, deep screening critically differs from classic display technologies in that such biases only manifest themselves in a single experiment, while biases (such as expression, folding, phage / yeast viability, non-specific binding (“stickiness”)) add up iteratively over multiple selection rounds.

We have expanded the discussion of how deep screening is positioned in the overall protein display space and how it relates to extant bulk solution selection approaches. We have adjusted the text to emphasise that classical selection methods are in general less efficient in enriching clones, and that library diversities of 10¹¹ are not necessarily required. Indeed, our data indicates that, at least for affinity maturation, repertoires of <10⁷ (including redundancies) are sufficiently functional provided they can efficiently mined as with deep screening. We saw this in particular with the IL-7 experiments, where picomolar affinity binders were identified from a library of <2x10⁵ clones, which is generally considered to be too small and where indeed our collaborators at AZ

were unable to identify any hits by phage display (personal communication). We have now also expanded this argument within the revised manuscript for increased clarity, although we do not refer to the negative results of our collaborators.

1.2.2. Furthermore, although deep screening is able to interrogate on the order of 10⁸ variants, this is still only a fraction of the possible sequence space. Therefore, authors should revise the manuscript text to describe deep screening in the context of the overall theoretical protein sequence space.

Again reviewer 1 makes a valid point. Theoretical protein sequence space is indeed vast and there are no methods to screen it comprehensively. However, in the context of antibody discovery and affinity maturation it should be noted that a more relevant term might be the sequence / shape space explored by mammalian immune systems, which are demonstrably sufficient to discover high-affinity binders to all (or most) antigens. The mouse B-cell repertoire is 10⁷ defining a minimal size of the primary antibody repertoire needed for universal *de novo* antibody discovery from a naïve library, while clonal B-cell diversity during affinity maturation is likely to be an order of magnitude lower. In this context deep screening reaches the screening depth needed for high-affinity antibody discovery. We have now expanded the discussion section to reflect these arguments in the revised manuscript.

1.3. In the proof-of-concept screen on the yeast-surface display nanobody (VHH) library, 2-fold over library background is considered a positive signal. Why is above this threshold considered bona fide HEL binding? Also, it is not clear what the library background is; it would be helpful to state more explicitly what this means.

We thank the reviewer for raising these questions and for highlighting our poor explanation. In deep screening we seek to discover binders by incubating the flow cell at different antigen concentrations and recording the fluorescent signal of individual clusters at these concentrations. At higher antigen concentrations background binding signal increases. We therefore define background signal as the fluorescent signal for all known clusters, and *bona fide* antigen binding by displayed VHH domains in relation to the background signal. Given systematic signal variations, we have found empirically that a 1.5-fold threshold reliably identifies true binders, even though an even less stringent threshold might be sufficient. As the VHH experiment was our first test run of the deep screening technology, we had not yet empirically defined a threshold and somewhat arbitrarily picked 2x above background as a conservative threshold. We have reworked the main text in the revised manuscript to provide a better explanation including a definition of 'library background'.

1.4. The authors present several correlations without the significance clearly stated, making data interpretation challenging. It would be helpful if the authors added p-values to these plots and mentioned in the text whether the correlations were statistically significant or just a trend.

We have now performed statistical analyses of all the reported correlations for antibody–antigen binding and have added p-values to all correlation plots in the revised manuscript.

1.5. The authors suggest at several points in the manuscript that the results generated via deep screening are precise enough to accurately rank clones by their binding ability. However, it is unclear how effectively deep screening is able to do this. For example, while there is a relatively strong correlation between deep screening binding and BLI-measured binding in figure S2B, this trend appears to be largely driven by the negatives. If the negatives were removed from this plot, there would likely be minimal correlation between the deep screening- and BLI-determined affinities. Consistent with this notion, in figure S11A, deep screening data provided low resolution with respect to differences in binding between the group of high-affinity clones (ML3-9, H3B1, and B1D2+A1) and between the group of low-affinity clones (G98A and C6.5). This limited ability of deep screening to accurately rank clones by their binding ability may underlie the relatively poor performance (F1-scores) of their BERT-DS model in identifying “low-hit” and “high-hit” sequences. Thus, while the authors seem correct in their assertion that deep screening can effectively separate binders from non-binders and identify high-affinity clones, it is not clear whether deep screening offers a high enough level of resolution to faithfully rank clones by binding affinity. This will require the authors to revise their statements / conclusions to be more accurate based on the actual data.

Reviewer 1 raises an important point. The fluorescent signal generated by antibody clusters is of course modulated by a range of factors beyond antigen binding affinities and kinetics including individual RNA clustering efficiency, antibody expression, folding etc. All of these factors contribute to imprecisions in ranking based on deep screening data and one or several of these factors may be at play in the Her2 affinity panel. Nevertheless, we note that even in this case no low affinity binders are assigned as high affinity (or vice versa, but rather that finer ranking within those broader categories K_D^{app} values are off. In this case, we believe this reflects limitations imposed by the antigen concentrations used in the equilibrium binding phase of the assay. Specifically, it is challenging to accurately fit a curve to clones G98A and C6.5, when neither of them reaches saturation. In the case of ML3-9, H3B1 and B1D2+A1 we observe a poor dynamic range from fitting a K_D^{app} , presumably due to poor expression or folding. Looking at K_D^{app} or fluorescent signal at any one concentration in isolation does not guarantee a 100% successful fine ranking, and we acknowledge this to be a limitation of the deep screening method in its current form.

However, we would like to point reviewer 1 to the anti-IL7 scFv data, where we observe a strong Spearman’s rank correlation ($r_s = -0.788$ at 0.3 nM IL7) and accurate ranking of clones, as demonstrated in Fig. 3E and Supplementary Fig. 7B – even when constructs were converted from scFvs to Fabs, expressed in CHO cells and purified. With respect to the Her2 affmat and the ML vs random libraries, we similarly observed a good correlation between BLI determined K_D s and deep screening determined fluorescent intensities at 3.3 and 10 nM Her2 and fitted K_D^{app} values. Specifically, Spearman’s rank correlations (r_s) of -0.612 for FI values and 0.563 for K_D^{app} values as shown in the new Supplementary Fig 13D and 13E in the revised manuscript.

Therefore, we believe it is justified to continue to claim that in general correlation between scFv deep screening data and “true” Fab affinities is strong and an overall ranking is accurate (even if a fine grained distinction is currently not possible). We note that despite these shortcomings the quality of our data was sufficient to enable successful training of a BERT model to outperform random mutagenesis by up to 35-fold and enable isolation of a 60pM K_D antibody to Her2.

1.6. The explanation of the BERT-DS machine learning model is inaccessible for a non-expert. The authors should explain what the model is and why it was selected for this particular task to frame the work. Why did the authors not consider other machine learning models / approaches?

We apologise for not explaining our reasoning clearer with regards to the machine learning model choice. We have now expanded the explanation of the BERT-DS model and rewritten it to be more accessible to a non-expert. We now also explain why the model was selected including an expanded number of references. We had previously only briefly mentioned that we had explored other machine learning models and found these to poorly predict beyond the training dataset. As suggested by yourself and reviewer 2, we have now also included a comparison and evaluation (ablation study) of our model +/- pretraining and the use of a soft classification target vs. 5 alternative machine learning models and present the new data in the revised manuscript in Supplementary Tables 1 -13 and a new Supplementary Fig. 19

1.7. Evaluating the pre-trained BERT-DS model by comparing its ability to predict sequences with better affinity than those in the “HER2affmat” dataset is a relatively low bar given that simple random mutagenesis and screening can complete the same task. Perhaps comparing the performance of the pre-trained BERT-DS model with a model that was not pre-trained, and/or a more simplistic model would more effectively demonstrate the predictive power and value of the pre-trained BERT-DS model.

We would like to respectfully disagree with this statement. While random mutagenesis can indeed also enable the isolation of high affinity binders, these are only present at a short mutational distance from the parent clones. There is therefore no way of knowing a priori if high affinity binders can be obtained in any arbitrary affinity maturation campaign using random mutagenesis. Furthermore, the BERT-DS model generated overall 5.7x more binders than random mutagenesis, which would provide greater opportunity of success in a drug discovery campaign. In some ways the efficiency of deep screening here obscures the inefficiencies of random mutagenesis as it allows us to discover the rare hits in the random mutagenesis library, which a less sensitive or lower throughput method may have missed. However, we agree that comparing BERT-DS in a pre-trained and non-pretrained form is an important test and we have performed this as well as a comparison / ablation analysis of our model with alternative machine learning models (see 1.6, 1.8) and present the new data in the revised manuscript in Supplementary Tables 1 -13 and a new Supplementary Fig. 19.

1.8. While the performance of BERT-DS is impressive, the authors didn't show any comparisons to more basic, conventional ML models. Is the BERT model necessary to achieve good predictions in a classifier?

We agree that this comparison is lacking from the paper. While we sought to minimise an excessive focus on ML models in the paper, we have undertaken a comparison and evaluation (ablation analysis) of our model +/- pretraining and the use of a soft classification target vs. 5 alternative, more conventional machine learning models and present the new data in Supplementary Tables 1 -13 and a new Supplementary Fig. 19. We show that classical ML models had a general failure to predict high-hit binders, or suffered from overfitting problems, while a MLP neural network yielded poor F1 scores when predicting high-hits.

1.9. Can the authors explain why they trained BERT-DS on the full VH dataset available on OAS, rather than human only sequences? As presumably those would allow the model to design much more relevant, therapeutic candidates.

We apologise for this omission. We filtered the OAS VH dataset to include only human sequences and pretrained on only 10% of the human OAS database for computing time reasons but did not explicitly state this in the paper. We have amended in the revised manuscript to clarify this point.

1.10. The authors show a PCA analysis of the CDR H3 sequence space for the HER2affmat library in Fig 4C. How the PCA analysis was done is not explained in the methods section.

We apologise for this omission. We now include a description of PCA analysis methodology in the methods section of the revised manuscript.

1.11. Choosing threshold for classifier is an essential step in the BERT-DS model set up. The author mentioned the threshold for high-hit was adjusted in order to have enough high affinity clones, while the ratio of high-hit sequence is still much lower than the other two classes. It will be helpful for others who are interested in implementing this workflow to know how the authors define “enough clone counts” and “balanced class counts”.

Thank you for raising this issue. We have adjusted text in the revised manuscript to better explain how and why this process was performed.

Reviewer #2 (Report for the authors (Required)):

The advantage of this platform is that in a single three day experiment the authors get a pseudo-binary readout of folding/expression/binding (mean FL1; more on this later) as well as sequences for hundreds of thousands of variants. This system affording sequence-function at scale represents a clear advantage for directed evolution affinity maturation campaigns. There are some stones to throw - the initial demonstration of the technology for initial binding discrimination from a library may not represent an advantage compared with further screening rounds by phage/yeast display; the authors, like many contemporaries, use deep learning models when simpler linear models would in fact be superior in reaching the desired answer; the system currently requires single chain or single domain antibodies; the dissociation rate measurements appear poorly correlated with functional antibody properties like dissociation constants. Still, these are manageable concerns and my overall enthusiasm for this paper is high. Given these critiques, the following should be addressed in any revision:

Major issues:

2.1.1 For the VHH anti-HEL screening experiment, there is some concern about the correlations reported. In Fig S2, dissociation constants for 25/41 (61%) of clones tested have a BLI-determined Kd of 10 μ M; presumably this is the upper limit of binding - based on the BLI curves presented these appear as non-binders and should be reported as such. I think the appropriate experiment here (can deep screening help identify binders better than randomly picking 96 colonies) should be tested with a binary classification scheme, and statistical significance reported (does deep screening do a better job of hit discrimination than simply plating colonies and screening?). I'm also not positive how to handle the non-binders in the libraries for evaluation of the Spearman Rho - panels 2B, C, but alternative ways of presenting this data should be considered by the authors.

This is a valid point and we accept that this is poorly explained in the text. We have now amended this in the revised manuscript. The upper limit of detection with the HiSeq 2500 is closer to 500 nM, although we have not thoroughly characterised this as it varies depending on the binding kinetics (on/off rates), protein displayed and the biophysical and labelling properties of the specific antigens. There is, of course, a legitimate concern about the poor transfer of BLI detected binding from deep screening identified and colony picked clones in the VHH experiment. A number of reasons may be in play for the poorer correlation (compared to the scFv experiments). These include the change of hosts, whereby selections were performed in yeast, deep screened with an in vitro translation extract and expressed in the E. coli periplasm. Furthermore, selections were not controlled for streptavidin binders, which were removed as hits in BLI but not in deep screening. The experiment suggested by the reviewer of comparing deep screening against picking 96 colonies was performed and described in the manuscript. It demonstrated that deep screening was capable of finding more and rarer low nM binders compared to colony picking. We have created a binary classification scheme as suggested by the reviewer to improve reporting of the non-binders. This is presented in the revised manuscript as Supplementary Fig. S2D with a ROC AUC curve.

2.1.2 (minor related point) - I think the claim 'identification of the same clones by standard procedures would have required either multiple further rounds of selection and/or labour-, cost-

and time-intensive microplate screening of thousands of colonies.' is overstated - by FACS you barely have binders when the experiment was stopped, so of course an antibody engineer would have continued with further rounds of yeast screening until robust signals were present. Recommend removing the last phrase from the sentence.

Our intention was of course to show that deep screening allowed us to identify binders at a much earlier point, when indeed yeast display and FACS (or MACS) had not yet fully enriched binders. We accept that one could simply continue with yeast display and FACS (or MACS) selections for a few more rounds and would likely identify binders, but clearly this would take time and cost some money after which microplate screening would still be needed to clonally identify binders. Nevertheless we have now rephrased this section in the text of the revised manuscript.

2.2. For the VHH anti-HEL screening experiment, dissociation constants do a poor job of describing or correlating with the data. From an outside perspective, the Illumina cell is ill-suited for BLI-type measurements as the flow cell can't remove antigen fast enough so you likely have some pseudo-equilibrium dependent on the local environment (antigen can unbind locally and rebind; the extent of local rebinding would be dependent on the presence of strong binder clusters nearby), so the data would be hard to deconvolute. My recommendation is just to remove or move all discussion of this to the supporting information (I understand a ton of work was involved here but it does not help advance the narrative of the technology).

The reviewer raises a valid point, and this is an indeed an underexplored aspect of the deep screening method. While we have not investigated this in detail, it seems like that an element of local rebinding and competition between proximal clusters may influence apparent dissociation rates, although we average intensities over at least 12 clusters in different locations on the flow cell. Nevertheless, these kinetic effects likely add a source of noise and variability between experiments, especially if clone X were to appear in two libraries with different compositions of binders. However, we also notice a tendency for clusters to “capture” some material once bound (possibly within the hydrogel like structure of the cluster); requiring excessive amounts of wash time to return the flowcell signal back to baseline. We would be very happy to discuss our observations with the reviewer at a future date. As our understanding of these effects is indeed still incomplete, we now refrain from detailed discussion in the main manuscript and have moved more of the analysis and discussion to the Supplementary information.

2.3.1 Figure 5. The result from affinity maturation by deep learning is overstated. The low hit and high hit categories had high amounts of false positives (F1-scores were low). Furthermore, the comparison group would be confusing to specialists. A very common method for affinity maturation is shuffling or combinatorially sampling mutations from the templated sequence for hits. The three 'seed' sequences reported in Figure 5 (HER2000[3-5]) have 1-5 mutations from the starting variant at 8 positions - the combinatorial library ($2^7 \times 3 = 384$ variants) is quite smaller than either libraries tested and would have reached the exact same answer faster. Inclusion of these language models is en vogue, but the present contribution does not add much to the paper.

We would like to respectfully disagree with this argument. The low-hit and high-hit categories likely have high amounts of false positives because we have binned the categories and reported predictions without considering the continuous nature of the underlying data.

As for the diversity argument, we would like to respectfully disagree with the claim that a combinatorial library of 384 would have sampled the diversity and reached the “*exact same answer faster*”.

There is a common misunderstanding on the combinatorial nature of diversity for the comprehensive sampling of mutations from hits resulting from a selection. In our case, we chose three of the best scoring mutants, termed ‘seed’ sequences and made between 1 - 5 mutations in each of these over 19 amino acid positions. Therefore, a combinatorial library for this sequence is defined by $(19 \text{ choose } n) * 20^n$, where n is number of mutations to be made. When applied over our mutational space, we obtain the following number of variants per seed:

$(19 \text{ choose } 1) * 20^1 = 380$ variants

$(19 \text{ choose } 2) * 20^2 = 68,400$ variants

$(19 \text{ choose } 3) * 20^3 = 7,752,000$ variants (7.75×10^6)

$(19 \text{ choose } 4) * 20^4 = 620,160,000$ variants (6.2×10^8)

$(19 \text{ choose } 5) * 20^5 = 37,209,600,000$ variants (3.72×10^{10})

Total = 3.783×10^{10} variants per seed.

Or 1.135×10^{11} variants for all three seeds.

The numbers remain extremely large if the positional space were restricted from 19 to 8 positions as suggested by the reviewer (total of 1.9×10^8 variants per seed). Therefore, the sequence space is not 384 variants, unless sampling is restricted to only single or null mutations

While it is true that our best scoring mutants in this experiment were found only two mutations away from their respective seed sequences, this still represents a combinatorial space of 68,400 variants per seed (and of $>2 \times 10^5$ for all 3 seeds). Sampling the entirety of this space would require 179 x 384 well plates per seed. Not to mention the logistical challenge of the experimental setup and assay design. Ordering the defined variant library in a bulk solution, then isolating clones into a single well at this scale is a non-trivial task and would almost certainly require both a robotic setup and oversampling at least 3x to obtain full coverage. This would bring the required number of 384 well plates to 537 per seed. Assuming assays are performed in a well volume of 10 μ L at 100 nM of Her2-biotin, one would require >1 L of Her2-biotin at 100 nM. Considering 200 μ g of Her2-biotin costs us £1,330 and you would require 7.83 mg to conduct this 537x 384 well screening experiment. This would consume $>£$ 52,000 ($>$ \$ 64,000) worth of commercially available Her2-biotin. In other words, sampling even just a complete set of double mutants over this space is impractical via classical microtiter plate screening, yet trivial by deep screening.

While in this case our best scoring mutants were found two mutations away from their respective seed sequences, this does not hold in general and may clearly depend on the particular antibody / antigen combinations as well as the final affinity sought, where sampling a wider variant space and greater mutational distances may be needed. The exploration of a wider sequence space around initial hits has obvious potential benefits beyond potential affinity increases including the generation of a larger and more divergent set of high-affinity leads for further development. Indeed, what we really sought to demonstrate using the BERT-DS model is that it can potentiate discovery of an expanded number of divergent high affinity hits by greatly reducing the amount of sequence space that would otherwise need to be (mostly unproductively) sampled (see, Supplementary Figure S16 and Table S3 showing a 7.9-fold improvement in yielding a high affinity hit over randomly on any two double mutants). In this context, it is highly significant that BERT DS provides a predictive performance advantage over random mutation sampling, and the performance gap scales with increasing mutation distance from the seed (to up 35-fold, see Supplementary Fig. 18).

2.3.2 If the authors insist on keeping this section, then very strong disclaimers must be added in results in discussion to the extent that one hot vector encoding of hits would have reached the same answer much faster and with minimal effort.

As to the reviewer's claim that a one hot vector encoding of the hits would have reached the same answer much faster and with minimal effort. We are unsure as to what the reviewer is referring to. A 'one hot vector encoding' is just an encoding scheme, but what is being done with the encoded data? Does the reviewer envision the use of a fully connected or convolutional network for classification?

To explore these arguments, we have now added a comparison of BERT-DS against various alternative machine learning models that use an embedding layer and / or a one hot vector encoding scheme to the revised manuscript. Our analysis shows a large degree of overfitting by these simpler models, as demonstrated by high training set performance and low test set performance. We also show that a logistic regression, linear SVM and random forest models are generally incapable of making accurate predictions over all three classes, as evident in their very low F1 scores. Furthermore, our analysis shows that a MLP significantly outperforms the classical models, but its performance is only about half as good as that of the BERT-DS model. Finally, we also tested the performance of BERT-DS with and without pre-training on OAS and observed a reduction in prediction performance in the absence of pre-training (as expected). These results and analysis are now presented in the new ablation study and Supplementary Tables S4-13 and Supplementary Fig. S19 in the revised manuscript.

We are open to continue this discussion and if reviewer 2 has other suggestions on how to reach a similar or superior performance to BERT-DS with minimal effort, we would be interested to explore such ideas.

Minor issues:

2.4.1 Abstract: The practical, demonstrated (not theoretical size limits) should be reported; in the supporting info authors discuss the demonstrated upper limit of 4e6 variants (supp page 3), which should be reported.

We have included the demonstrated diversity in the abstract

2.4.2. Introduction: The introduction does a reasonable job of surveying the literature; one paper missing is from the Cochran group and should be included and compared with this work: Chen, Bob, et al. "High-throughput analysis and protein engineering using microcapillary arrays." Nature chemical biology 12.2 (2016): 76-81.

Many thanks for pointing out this interesting paper, which we had indeed missed. Now included.

2.4.3. Reporting of error. For Figure 2: —> Meaning of error bars and number of replicates should be described in the figure legend. For Figure 3. —> 'error bars are 1 s.d., n=2'. It is impossible to record a standard deviation from two data points. Please clarify.

Thank you for pointing this out to us. We have corrected Figures 2 and 3 accordingly.

2.4.4. For Figure 3, the panel C uses DS FI at 333 pM, but main text states clones were selected by top 19 binding at 1 nM hUIL-7. Please clarify or show the selection data. For the figure it may be appropriate to show a volcano or other plot showing the distribution of DS FI values among clones to show clone selection.

Thank you for pointing this out. We show 333 pM in this panel as it demonstrates the ability to rank clones, while at 1 nM of hUIL-7, we can only see binders in a binary manner. We have added a rank plot to the SI for all measured datapoints for the selected clones and for the entire library distribution, shown in Supplementary Figs 7, 13 in the revised manuscript.

2.4.5. Supporting info. The equations listed in the supplemental should be updated to reflect concordance with the main text (in particular KD_{app} and $k_{off,app}$) (page 9; a biexponential fit is performed without elaboration as far as I can ascertain).

Thank you again for alerting us to these errors. We have updated these to reflect the main text and explained why we use a biexponential fit.

Reviewer #3 (Report for the authors (Required)):

3.1 In this paper, Porebski et al. provide a platform for fast discovery of binders. While I am sure that speed in discovery is a valuable asset, it is unclear to me how this platform fundamentally differs from other library discovery methods.

The deep screening platform differs from phage / yeast / ribosome display methods in that it does not require iterative rounds of panning (with their inherent and cumulative biases). Furthermore, instead of just isolating “winners” as in repertoire selection, deep screening provides a real-time, global and internally consistent survey of binding activities within the library including non-binding (or weakly binding) clones. The latter is a crucial advantage of deep screening datasets for the training of machine learning models.

Deep screening differs from other direct screening approaches like e.g. robotic microplate screening or ProtMaP on the MiSeq platform by the covalent linkage of the mRNA to the flowcell surface improving display performance and enabling stringent washing steps, by its ~100-fold enhanced screening depth (compared to ProtMaP). We have rewritten several sections of the revised manuscript to outline the above arguments more clearly.

3.2. The machine learning section reads very antiquated not based on the state-of-the-art. This is probably because the authors go out of their way to almost not cite any paper of relevance from the last years (both from the experimental and computation side of antibody discover).

We would like to respectfully disagree. Large language models pre-trained to solve masked amino acid or next amino acid prediction on large protein sequence datasets before being fine-tuned on application specific datasets are entirely state of the art. Since the development of this work, larger models such as ESM v2 from Facebook have been made publicly available, however, they add no specific value to antibody affinity prediction as this information is not conserved in large sequence datasets. Diffusion model approaches, such as RFdiffusion are a viable approach for de novo binder design, but do not yet appear to provide a strategy for conditioning or refinement using a library-antigen specific dataset.

If the reviewer has concrete suggestions as to the state of the art that should be included or on how we could improve our machine learning model performance, we would be happy to include this in our discussion.

As for paper citations, we have sought to cite the as many of the most relevant papers in the literature as relevant to our approach as possible within journal limits. However, we are aware that the body of literature in this field is vast and we apologize if we have missed out manuscripts that reviewer 3 considers important. To remedy this, we now provide a link to an online database (compiled by Kevin K. Yang of Microsoft Research (<https://github.com/yangkky/Machine-learning-for-proteins>)) which enumerates a large number of papers in the field with special focus on applications in protein engineering as part of the Supplementary information. Again, we would value the reviewer’s suggestions of especially important manuscripts that we may have missed and would be happy to include these.

3.3 The authors show a lot of data but these data do not advance our understanding of library methods nor antibody biology.

We would like to respectfully disagree. With regards to library methods, deep screening for the first time provides global estimates of the phenotypic richness and functional distribution in large synthetic antibody libraries and provide the tools for making these more functional in the future. It also suggests that BERT transformer models can be useful to productively assist library construction and mutant design. Finally, as briefly discussed in 1.2, the interplay between repertoire diversity (as generated by the immune system or synthetically by random, semi-random or defined mutation) and antibody function is a key aspect of antibody biology and immune system function. Although a detailed discussion of these aspects is beyond the present manuscript, deep screening promises to be a powerful tool to study it.

Rebuttal 2

Reviewer #1

The authors have sufficiently responded to most of my comments and the revised manuscript has been improved. However there are still two remaining issues related to the following:

1.8 response

The difference in F1 between the pretrained vs. random initialization BERTs trained in 2023 is less than the F1 between pretrained 2023 vs pretrained 2021 models. How much of the difference in performance seen between pre-train and random initialization might be due to train/test set batching? With so much variability, are the authors able to be so certain that pretraining contributes significantly to the performance of the DS-BERT model?

1.11 response

The authors reformatted the classifier structure, however still missing the illustration of why choosing the threshold which yielded 111 high-hit was enough for training and a class count of 232,693 non-hit, 1,284 low-hit and 111 high-hit was considered to be balanced.

We suspect that the difference in F1 between the pre-trained model in 2023 versus 2021 is potentially due to differences in software and hardware used for training. This would have resulted in different initialization values, even though random seeds were kept constant. Given the variability on retraining, we cannot reliably compare the 2021 model against the random initialized 2023 model. In the SI ablation study discussion, we comment on this, and state that we need to only consider the pretrained 2023 model in evaluating whether pre-training results in benefit. Variability is a pervasive and often undisclosed topic in many machine learning publications, so we appreciate the reviewer's skepticism and concerns. We are happy to include additional statements that balance the certainty.

It is also worth noting that OAS predominantly contains human antibodies determined through immune repertoire sequencing. Given that Her2 is a human protein, it is entirely plausible that OAS is biased against antibodies that bind to self-proteins, and thus does not add the same benefit as if we were using a non-self antigen. We certainly think this is worthy of a future study, as well as investigating the use of different pre-training datasets, and alternate models such as ESM.

With respect to the classifier structure, thank you for catching this omission and lack of detail. We have amended the main text and SI to explain how these thresholds were chosen and updated the language around balance of the classes.

Reviewer #2

The authors have done a fantastic job of responding to most of my concerns in the initial manuscript. Let me reiterate that the platform development and execution is top-rate and there are no technical concerns with the development of the platform, the demonstrations of the technology, or the results presented.

With great respect to the authors, the remaining concern I have is with the overstatement of the DL model presented compared with what a typical protein engineer workflow would comprise for a task-specific engineering exercise (improve the properties of an antibody). Before I go in further detail, let me mention that I have no technical concerns with the ML portion of the paper and I commend the authors for inclusion of competing ML analysis (MLP, SVN, etc) - this strengthens the paper. So, on balance, I don't need to re-review this paper as it is technically sound and represents an impactful advance. Rather, I'll leave this to the editor's discretion (and, I suppose, the authors). I feel strongly that down-weighting the significance of the ML model is important, and this is where my subjectivity shows - while I am quite bullish on unsupervised learning to eventually learn functional properties of antibodies, my personal opinion is that many overstate the importance of such models for task-specific purposes (improve the properties of antibody 'A'), when simpler models reach the 'best' answer faster and with much less labour. Let me explain my reasoning.

It is well established in directed evolution of biomolecules that if the screen/selection is towards some objective function (higher yields, lower K_m , lower K_d , folding propensity, etc) that the hits resulting from the screen can be combined; this can be through DNA shuffling or through combinatorial library generation through synthetic DNA. The methods are immaterial. What you are doing in practice is sampling within the functional space of the molecules identified from the first round of screening. So, contrary to the claims from the authors response on the combinatorial complexity of sampling, the actual functional space you would seek to sample in practice is much lower. There is a good older literature reference for this in Nature Biotech (ref 1) - the crux of this is a very old supervised learning technique (a form of linear regression!) based on one hot encoding of amino acid positional hits (this is what I was trying to convey in my first review; the authors rightly pushed back as I was imprecise) was fantastic at navigating functional search space.

In this particular example, the authors identified initial hits as seed sequences comprising the following sets of mutations:

98GA
99YQW
101SDT
102ST
103SP
104NRT
106AL
107KT

So, faced with these hits, a protein engineer would encode the combinatorial library encoding diversity at these positions, these positions only, and only these residues. If she/he did, they would end up with the exact same answer (to the question: 'find the best antibody') as the authors did with a fraction of the effort (by only screening through several hundred sequences in this example):

98G

99Y
101S
102T
103S
104R
106L
107T

What if there were more hits than those represented in the seed sequence? In practice, the authors would convert the frequency at each position by one hot encoding and encode combinatorial libraries of the most frequently represented residues at each position in the CDR; since those seed sequences likely represent positions with high frequency in the hits, the answer would likely be contained in the encoded library, again with a fraction of the effort. It would have been fantastic if the language model used by the authors identified residues not present in the seed sequences for the best hit, but this was not the case.

I'm not quite sure how the authors can address this or what suggestions to convey to the editor - the language model used does give additional information (what are sequences not likely to bind the given target) that you can't get with this crude workflow, but for the simpler task of identifying the best antibody sequence other methods are superior in that they would identify the best answer as the language model but with a fraction of the effort. My original review suggested a strong rephrasing of the language stating the concerns noted above; I still believe this for the reasons discussed above. Ultimately, the novelty of the paper is in the assay, not the language model.

References:

*1. Fox, R. J., Davis, S. C., Mundorff, E. C., Newman, L. M., Gavrilovic, V., Ma, S. K., ... & Huisman, G. W. (2007). Improving catalytic function by ProSAR-driven enzyme evolution. *Nature biotechnology*, 25(3), 338-344.*

We welcome an opportunity to further discuss these points.

Reviewer 2 is correct in that the impact of ML models on (antibody) engineering outcomes should not be overstated and indeed, there are quite a number of examples in the literature where that has been the case. However, we would like to argue that (i) we do not overstate the impact in our case and (ii) we provide a fair comparison between outcomes using either ML-instructed library generation vs. random mutation.

Reviewer 2 is correct in that combinatorial repertoires constructed from amino acid preferences might have yielded similar outcomes. However, it is important to note that mutations are not universally additive and epistatic (non-additive) effects are arguably more likely within a loop where an interdependence of amino acids is likely to maintain loop conformation.

Our main motivation to include the ML section in the manuscript was to demonstrate that deep screening datasets are of sufficiently high quality to be used for training of a language model. Given the size and complexity of these datasets it could not be priori assumed that they could be used as a learning input. Indeed, using antibody affinity/sequence datasets generated by academic and commercial investigators, we had previously not observed any successful training outcomes, due to insufficient scale, indirect measurement of function, collection of data over multiple days resulting in assay drift, collection on different hardware, and or collection by multiple investigators (not shown).

The demonstration that deep screening datasets are viable inputs for transformer ML approaches (and less for other ML approaches as shown in the revised manuscript) is valuable because it opens up the analysis and learning of libraries with greater complexity and that are less focused on a particular region than explored here. The above approach of additive mutational sampling becomes increasingly impractical when the number of varied CDRs or sample space is increased.

We have amended the manuscript text to make these arguments more clearly.